# MICOS assembly controls mitochondrial inner membrane remodeling and crista junction redistribution to mediate cristae formation

Till Stephan[1,2,†] , Christian Brüser[1,2,†], Markus Deckers[3], Anna M Steyer[4], Francisco Balzarotti[1], Mariam Barbot[1,2], Tiana S Behr[1,2], Gudrun Heim[5], Wolfgang Hübner[6], Peter Ilgen[1,2], Felix Lange[1,2], David Pacheu-Grau[3], Jasmin K Pape[1] , Stefan Stoldt[1,2], Thomas Huser[6], Stefan W Hell[1,7], Wiebke Möbius[4] , Peter Rehling[3,7] , Dietmar Riedel[5] & Stefan Jakobs[1,2,7,*]

## Abstract

Mitochondrial function is critically dependent on the folding of the mitochondrial inner membrane into cristae; indeed, numerous human diseases are associated with aberrant crista morphologies. With the MICOS complex, OPA1 and the $F_1F_o$-ATP synthase, key players of cristae biogenesis have been identified, yet their interplay is poorly understood. Harnessing super-resolution light and 3D electron microscopy, we dissect the roles of these proteins in the formation of cristae in human mitochondria. We individually disrupted the genes of all seven MICOS subunits in human cells and re-expressed Mic10 or Mic60 in the respective knockout cell line. We demonstrate that assembly of the MICOS complex triggers remodeling of pre-existing unstructured cristae and de novo formation of crista junctions (CJs) on existing cristae. We show that the Mic60-subcomplex is sufficient for CJ formation, whereas the Mic10-subcomplex controls lamellar cristae biogenesis. OPA1 stabilizes tubular CJs and, along with the $F_1F_o$-ATP synthase, fine-tunes the positioning of the MICOS complex and CJs. We propose a new model of cristae formation, involving the coordinated remodeling of an unstructured crista precursor into multiple lamellar cristae.

**Keywords** cristae biogenesis; electron microscopy; MINFLUX; nanoscopy; super-resolution microscopy

**Subject Categories** Membrane & Trafficking; Organelles

The EMBO Journal (2020) 39: e104105

See also: **K McArthur & MT Ryan** (July 2020)

## Introduction

Mitochondria are essential organelles of eukaryotic cells that perform a multitude of functions. Most notably, they are the powerhouses of the cell that generate ATP through oxidative phosphorylation. Mitochondria feature two membranes, the smooth outer membrane (OM) and the highly convoluted inner membrane (IM). The latter is functionally and structurally divided into two domains, namely the inner boundary membrane (IBM) that parallels the OM and crista membranes (CMs), infoldings of the IM pointing toward the interior of the organelle. The CMs are connected to the IBM by small circular to slit-like openings which are called crista junctions (CJs). The proper folding of the IM is closely related to the function of the organelle and numerous devastating diseases, including cardiomyopathies, neurodegenerative disorders, metabolic diseases, and cancers are associated with aberrant CM folding (Chan, 2012; Nunnari & Suomalainen, 2012; Friedman & Nunnari, 2014; Pernas & Scorrano, 2016; Wai & Langer, 2016).

In mammalian cells, regularly spaced lamellar cristae appear to be the dominant IM morphology, while also other cristae shapes exist (Hackenbrock, 1966; Scheffler, 2008; Cogliati et al, 2016). Although the basic membrane architecture of mitochondria was discovered almost seven decades ago (Palade, 1952; Sjöstrand, 1953), the molecular mechanisms underlying the formation of cristae are still poorly understood and several models of cristae biogenesis and maintenance have been suggested (Rabl et al, 2009; Zick et al, 2009; Harner et al, 2016; Muhleip et al, 2016). Recent experimental evidence supports the hypothesis that in budding yeast, lamellar cristae are generated by the conversion of two IBM sheets following a mitochondrial fusion event (Harner et al, 2016).

1   Department of NanoBiophotonics, Max Planck Institute for Biophysical Chemistry, Göttingen, Germany
2   Clinic of Neurology, University Medical Center Göttingen, Göttingen, Germany
3   Department of Cellular Biochemistry, University Medical Center Göttingen, Göttingen, Germany
4   Department of Neurogenetics, Electron Microscopy Core Unit, Max Planck Institute of Experimental Medicine, Göttingen, Germany
5   Laboratory of Electron Microscopy, Max Planck Institute for Biophysical Chemistry, Göttingen, Germany
6   Department of Physics, University Bielefeld, Bielefeld, Germany
7   Cluster of Excellence "Multiscale Bioimaging: from Molecular Machines to Networks of Excitable Cells" (MBExC), University of Goettingen, Goettingen, Germany
    *Corresponding author. Tel: +49 (0) 551 2012531; E-mail: sjakobs@gwdg.de
    †These authors contributed equally to this work

In this study, we show that in human cells, lamellar cristae generation does not require mitochondrial fusion, suggesting fundamental difference in the cristae biogenesis pathways of budding yeast and higher eukaryotes.

Three membrane-shaping factors, namely the dimeric $F_1F_o$-ATP synthase, the MItochondrial contact site and Cristae Organizing System (MICOS), and the large GTPase optic atrophy 1 (OPA1), exhibit crucial, yet different roles in cristae biogenesis and maintenance. The $F_1F_o$-ATP synthase associates into elongated rows of dimers that introduce positive curvature into membranes and stabilize the rims of cristae. Thereby, these dimer rows are important determinants of cristae morphology (Strauss *et al*, 2008; Davies *et al*, 2012; Muhleip *et al*, 2016; Blum *et al*, 2019). In addition, remodeling of the IM is critically influenced by OPA1 (Mgm1 in yeast) (Alexander *et al*, 2000; Delettre *et al*, 2000; Zanna *et al*, 2008; Varanita *et al*, 2015; MacVicar & Langer, 2016). OPA1 has a role in IM fusion (Cipolat *et al*, 2004; Ishihara *et al*, 2006) and is presumably also involved in fission (Anand *et al*, 2014). It also influences the overall cristae architecture and controls the CJ diameter in apoptosis (Frezza *et al*, 2006; Meeusen *et al*, 2006). Recently, it was shown that Mgm1 from *Chaetomium thermophilum* can assemble into a helical filament on positively and negatively curved membranes, leading to the proposal that Mgm1 might form a helical filament inside of CJs (Faelber *et al*, 2019). MICOS is a large, hetero-oligomeric protein complex that is primarily located at CJs (Harner *et al*, 2011; Hoppins *et al*, 2011; von der Malsburg *et al*, 2011; Alkhaja *et al*, 2012; Rampelt *et al*, 2017b). In humans, seven MICOS subunits have been identified so far, namely Mic60 (Mitofilin), Mic27 (APOOL), Mic26 (APOO), Mic25 (CHCHD6), Mic19 (CHCHD3), Mic13 (QIL1), and Mic10 (MINOS1) (Pfanner *et al*, 2014; van der Laan *et al*, 2016). The holo-MICOS complex consists of two distinct subcomplexes (Mic10/13/26/27 and Mic60/19/25) that were named according to the core components Mic10 and Mic60, respectively (Friedman *et al*, 2015; Guarani *et al*, 2015; Anand *et al*, 2016). Both, Mic10 and Mic60 show membrane-shaping activity (Barbot *et al*, 2015; Bohnert *et al*, 2015; Hessenberger *et al*, 2017; Tarasenko *et al*, 2017). Nevertheless, the exact functions of the two MICOS subcomplexes are unknown. The depletion of several MICOS subunits causes a depletion of CJs and the formation of detached CMs in mitochondria (Harner *et al*, 2011; Hoppins *et al*, 2011; von der Malsburg *et al*, 2011).

Genetic and physical interactions between the $F_1F_o$-ATP synthase, OPA1, and MICOS have been demonstrated (Rabl *et al*, 2009; Darshi *et al*, 2011; Janer *et al*, 2016; Eydt *et al*, 2017; Rampelt *et al*, 2017a; Quintana-Cabrera *et al*, 2018), but the functional interplay of these three major players involved in cristae development remained largely elusive.

In this study, we used CRISPR/Cas9 genome editing to individually disrupt the genes of all seven MICOS subunits. We demonstrate that the Mic60-subcomplex controls the formation of CJs, whereas the Mic10-subcomplex is crucial for the formation of lamellar cristae. Inducible stable cell lines allowed us to follow the restoration of lamellar cristae upon re-expression of MICOS proteins. We found that re-formation of the holo-MICOS complex caused extensive remodeling of pre-existing aberrant cristae, including also the formation of secondary CJs. We further demonstrate that OPA1, next to stabilizing tubular CJs, influences along with the $F_1F_o$-ATP synthase the positioning of the MICOS complex. Our findings suggest a new model of cristae formation, based on coordinated membrane remodeling of unstructured CMs into highly ordered cristae.

# Results

## HeLa cells feature primarily lamellar cristae

Live-cell 2D STED nanoscopy of HeLa cells stably expressing cytochrome *c* oxidase subunit 8A (COX8A) C-terminally fused with a SNAP-tag revealed that these cells predominantly exhibit groups of lamellar cristae spaced by voids that are occupied by mitochondrial nucleoids (Fig 1A and C) (Stephan *et al*, 2019). To further investigate the fold of the IM in three dimensions, we performed focused ion beam milling combined with scanning electron microscopy (FIB-SEM; Fig 1B; Appendix Fig S1, Movie EV1). Reconstructions of FIB-SEM data revealed a substantial level of structural heterogeneity of the architecture of the IM. Groups of stacked cristae were often rotated with respect to each other. Occasionally, we observed single cristae oriented perpendicular to the longitudinal axis of the mitochondrion and also a twisted arrangement of the cristae as previously described in mitochondria from yeast (Stoldt *et al*, 2019) and flies (Jiang *et al*, 2017) (Movie EV1). Mic60 has been demonstrated to form clusters that are enriched at CJs (Harner *et al*, 2011; Alkhaja *et al*, 2012; Jans *et al*, 2013). In HeLa cells, these clusters often resembled a stripe pattern perpendicular to the longitudinal axis of the mitochondria (Fig EV1A). Live-cell STED nanoscopy recordings of HeLa cells expressing Mic10-SNAP further verified an arrangement of Mic10 clusters in a perpendicular stripe pattern, suggesting that single lamellar cristae can exhibit multiple CJs around a mitochondrion (Movie EV2). Unexpectedly, Mic60 clusters also appeared at sites marked by the mitochondrial nucleoids that are usually free of cristae lamellae (Figs 1C and D, and EV1A) (Stephan *et al*, 2019). In order to verify the occurrence of Mic60 clusters in the absences of properly developed lamellar cristae, we first labeled cells for COX8A-SNAP and subsequently chemically fixed and immunolabeled them for Mic60. Dual-color STED recordings confirmed that while the majority of Mic60 is present at the cristae lamellae, Mic60 clusters also appear in areas devoid of lamellar cristae (Figs 1E and EV1B).

## Characterization of MICOS-KO cell lines

### Depletion of Mic60, but not of Mic10, results in the absence of all MICOS subunits

To dissect the role of the individual MICOS subunits, we disrupted the genes of the seven MICOS proteins (*MIC10, MIC13, MIC19, MIC25, MIC26, MIC27,* and *MIC60*) in HeLa cells utilizing CRISPR/Cas9 genome editing (for details on the experimental strategy, see Appendix Table S1). We analyzed the protein levels of all MICOS subunits in cell lysates and isolated mitochondria from the MICOS-knockout (KO) cell lines (Figs 2A and EV1C). Deletion of Mic60 was associated with an almost complete loss of Mic10, Mic13, Mic19, and Mic26 and a strong reduction of the Mic25 and Mic27 protein levels (Fig 2A). In Mic19-KO cells, we observed a similar reduction of all MICOS subunits. In contrast, the depletion of Mic10 was associated with a strong reduction of Mic13 and Mic26, whereas the

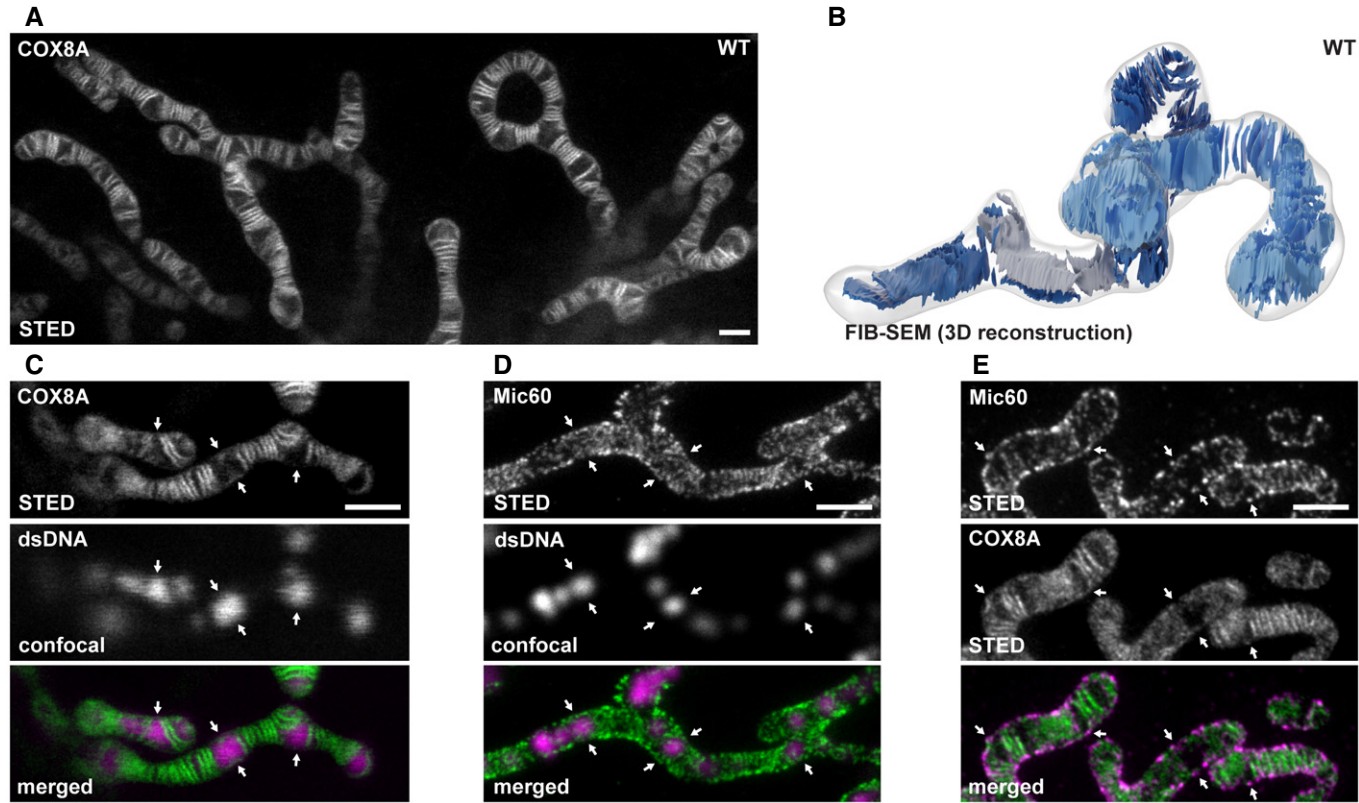

**Figure 1.   Inner membrane architecture and MICOS distribution in HeLa cells.**

A   Live-cell nanoscopy of HeLa cells. COX8A-SNAP was labeled with SNAP-cell SiR and imaged with STED nanoscopy.
B   3D architecture of cristae in HeLa cells visualized by FIB-SEM. The OM and the IBM are together shown as a transparent surface. Cristae of two mitochondria are depicted in light and dark blue. A twisted crista is shown in gray.
C   Live-cell recording as in (A). In addition, mitochondrial DNA was stained with PicoGreen and imaged by confocal laser scanning microscopy. Arrows mark sites where nucleoids but no lamellar cristae are present.
D   STED nanoscopy of mitochondria immunolabeled for Mic60 and dsDNA. Arrows mark sites where nucleoids are present.
E   Dual-color STED nanoscopy of cells labeled for COX8A-SNAP and immunolabeled for Mic60. Arrows mark Mic60 clusters in the absence of fully developed lamellar cristae.

Data information: Scale bars: 1 μm.

levels of Mic19, Mic25, Mic27, and Mic60 were unaltered. Similarly, the depletion of Mic13 was associated with a decrease in Mic10 and Mic26 levels (Fig 2A). In cell lysates, the levels of other mitochondrial proteins including TOM20 suggested a slightly reduced amount of mitochondria in Mic60-KO and Mic19-KO cells (Fig EV1C), whereas the other KO cell lines were unaffected.

As Mic60 controlled the abundance of all other MICOS proteins (Figs 2A and EV1C), we next performed complex immunoprecipitation (Co-IP) experiments from isolated mitochondria using Mic60 antibodies to investigate whether Mic60 interacts with the remaining MICOS subunits when specific subunits are depleted. In wild-type (WT) cells, we isolated Mic10, Mic13, Mic19, Mic25, Mic26, and Mic27, demonstrating a fully assembled holo-MICOS complex (Fig 2A). For the different KO cell lines, the amounts of isolated proteins largely coincided with the steady-state levels, suggesting that none of the subunits is essential for the binding of the remaining subunits to Mic60 (Fig 2A). As an exception, we found that in the absence of Mic10, little Mic27 was co-isolated with Mic60, although Mic27 was still present at relatively high levels in these cells.

In line with sequencing data, immunoblotting did not reveal any Mic60 either in whole-cell extracts or in isolated mitochondria from Mic60-KO cells (Figs 2A and EV1C, Appendix Table S1). Unexpectedly, after immunoprecipitation with Mic60 antiserum, we detected on Western blots a faint signal for Mic60 and Mic19 in the Mic60-KO cells (Fig 2A). Possibly, an unknown isoform or alternative splicing might account for the residual amount of Mic60 in these cells. As we could not identify any cell line without this residual expression of Mic60, it is currently unclear, if these trace amounts of Mic60 are functionally relevant.

Altogether, we conclude that the depletion of Mic60 leads to an almost entire loss of MICOS. In contrast, the lack of Mic10 causes specifically a depletion of the Mic10-subcomplex in human cells.

### Assembly of OXPHOS is largely unaffected in MICOS-KO cells

In order to analyze the influence of the loss of MICOS subunits on the assembly of the supercomplexes of the oxidative phosphorylation (OXPHOS) system, we performed blue native polyacrylamide gel electrophoresis (BN–PAGE) of mitochondria isolated from the MICOS-KO cells (Fig EV1D). BN–PAGE showed only slight

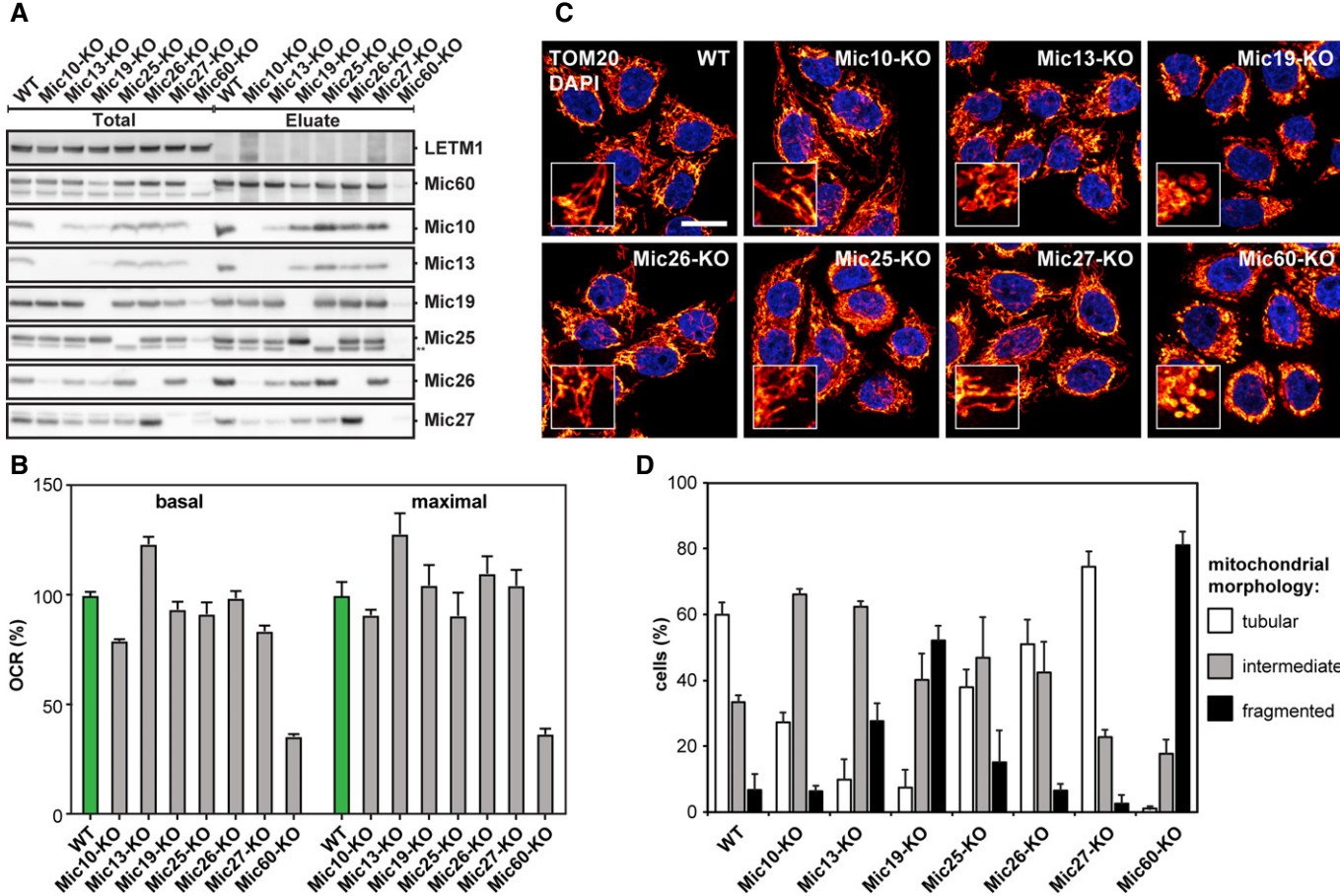

**Figure 2. Characterization of MICOS knockout cell lines.**

A   Composition of MICOS in WT cells and MICOS mutants. Mic60-specific antibodies were used to pull-down Mic60 and interacting proteins from insolated mitochondria. ** Unspecific band, due to the cross reaction of the anti-Mic25 antibody with Mic19.

B   Oxygen consumption rate (OCR) of WT cells and MICOS mutants as analyzed by Seahorse Analyzer. Basal and maximal OCR are shown relative to the WT. Error bars: SEM, $n = 6$.

C   Mitochondrial networks of WT and MICOS mutant cells. Cells were immunolabeled for TOM20 and visualized by confocal microscopy. The inset shows a magnification of the respective overview image. Scale bar: 10 µm.

D   Quantification of the mitochondrial networks as shown in (C). The evaluation was performed manually in a blinded approach based on pre-defined morphology criteria. Average and SD of three independent biological replicates are shown (> 170 cells per sample and replicate).

Source data are available online for this figure.

differences between the individual MICOS-KO cells and the WT. Even in the absence of Mic60, virtually resulting in the absence of MICOS, the assembly of complexes I, II, and V was only somewhat impaired and the assembly of complexes III and IV was slightly decreased. This was also apparent for the corresponding supercomplexes (Fig EV1D). In Mic60-KO cells, the oxygen consumption rate was reduced, but not abolished; all other MICOS-KO cell lines exhibited oxygen consumption rates close to the WT (Fig 2B). We conclude that the deletion of MICOS subunits has only modest influence on OXPHOS assembly.

### Morphology of the mitochondrial network primarily depends on the Mic60-subcomplex

Next, we analyzed the effects of the depletion of the MICOS subunits on the overall morphology of the mitochondrial network (Fig 2C

and D). Mic19-KO and Mic60-KO cells showed the most severe phenotype with strongly fragmented mitochondrial networks and large spherical mitochondria. Cells lacking Mic10, Mic13, Mic25, and Mic26 displayed on average only moderately fragmented mitochondrial networks, whereas the Mic27-KO cells exhibited slightly hyperfused mitochondrial networks (Fig 2C and D). These findings demonstrate that the Mic60 subcomplex, but not the Mic10 subcomplex, is crucial to maintain the mitochondrial network structure.

### Loss of Mic10, Mic13, Mic19, and Mic60 disturbs cristae architecture

To analyze the cristae morphology of the seven MICOS-KO cell lines, we first performed transmission electron microscopy (TEM; Fig 3A, Appendix Fig S2A). For an initial evaluation, we classified the structures of the cristae as "wild type" (ordered, lamellar

cristae) or "aberrant" (disordered cristae; Fig 3A and B). In Mic60-KO cells, virtually all mitochondria had an aberrant cristae morphology. Similarly, the Mic10-, 13-, and Mic19-KO cells showed a strong phenotype, as more than 75% of the mitochondria had aberrant cristae. The mitochondria of the Mic25-, Mic26-, and Mic27-KO cells exhibited only mild phenotypes with more than 75% of the mitochondria showing ordered lamellar cristae (Fig 3A and B). Additionally, RNA interference (RNAi) experiments independently confirmed these phenotypes (Appendix Fig S2B and C).

To further characterize the cristae shapes in the Mic10-, Mic13-, Mic19-, and Mic60-KO cells, we assigned the cristae phenotypes to more detailed categories. In Mic10 and Mic13-KO cells, we mostly observed a large single CM that paralleled the IBM (Fig 3A and C). This phenotype was less frequent in mitochondria from Mic60- or Mic19-KO cells, which instead regularly showed multiple layers of CMs arranged in stacks or onion-like arrangements (Fig 3A and C, Appendix Fig S2A).

### Tubular CJs exist in Mic10-KO cells but not in Mic60-KO cells

We estimated the number of CJs in each of the seven KO strains on several hundred TEM cross sections through mitochondrial tubules. For a comparison of the different cell lines, the number of CJs was normalized to the length of the OM (Fig 3D). Compared to WT cells, the occurrence of CJs was reduced by about 25% in Mic26-KO cells and by more than 70% in Mic10-, Mic13-, and Mic19-KO cells. In Mic60-KO cells, the number of CJs was close to zero. We note, however, that in Mic60-KO cells, we occasionally observed connections of the IBM with the CM (Fig 3D). WT cells, as well as Mic13-, Mic19-, Mic25-, Mic26-, and Mic27-KO cells, had CJs with an average diameter of about 20 nm (Fig 3E). In Mic10-KO cells, the average diameter was enlarged to about 28 nm ($n = 96$), and for the few unusually shaped connections found in Mic60-KO cells ($n = 26$), the average diameter was about 32 nm (Fig 3E).

Compared to previous findings in yeast (Harner *et al*, 2011; Barbot *et al*, 2015; Bohnert *et al*, 2015), the presence of a substantial number of CJs (about 20% of the WT) in Mic10-KO cells was unexpected. To further analyze this finding, we performed electron tomography (ET) of WT, Mic10-KO, and Mic60-KO cells (Fig 3F). Recordings of tilt series thereby allowed us to analyze and reconstruct the CMs and CJs within mitochondrial sections of approximately 200 nm thickness.

Tomograms of Mic60-KO mitochondria confirmed that the CMs frequently formed multilayered, very long sheets, resulting in an onion-like architecture. As expected from the TEM recordings, we found only very few CJs in the tomograms (Fig 3F, Movies EV3 and EV4). Vesicular structures in the Mic60-KO mitochondria usually proved to be tubular extensions of large sheet-like cristae (Fig 3F, Movie EV3).

The tomograms also confirmed that the lamellar cristae of WT cells exhibited numerous circular or slit-like CJs (Fig 3F; Movie EV5). CJs connected to the same single crista were often in close proximity and thereby they formed a line pattern perpendicular to the longitudinal axis of the mitochondrial tubule (Fig 3F; Movie EV5). As Mic60 is enriched at CJs (Jans *et al*, 2013), the distribution of the CJs explains the perpendicular stripe pattern seen in STED images of Mic60 in these cells (Figs 1D and EV1A).

The tomograms showed that in mitochondria of Mic10-KO cells, the cristae usually formed single-layered large sheets with circular, stalk-like tubular CJs (Fig 3F, Movie EV6). In the rare Mic10-KO mitochondria with onion-shaped IM morphology (Movie EV7), these CJs connected the outermost CM with the IBM (Fig 3F, Movie EV7). The stalk-like CJs found in Mic10-KO cells were structurally different to those observed in WT cells, which were circular or slit-like and immediately continuous with the lamellar cristae. Furthermore, in Mic10-KO cells, the CJs appeared to be irregularly distributed and, compared to the WT, often confined to small areas.

Taken together, Mic10-, Mic13-, Mic19-, and Mic60-KO cells show a strongly altered cristae architecture. Remarkably, human mitochondria depleted of Mic10 still exhibit numerous tubular, stalk-like, slightly enlarged CJs, whereas CJs are nearly absent in Mic60-KO cells. As Mic10-KO cells accommodate the Mic60-subcomplex, we conclude that the Mic60-subcomplex, but not the Mic10-subcomplex, is necessary for the formation of CJs.

### Mic10 controls the spatial distribution of Mic60 and the formation of Mic60 assemblies

The irregular distribution of CJs in Mic10-KO cells suggested that in these cells also the Mic60-subcomplexes exhibit an aberrant distribution. To test this, we performed 2D STED nanoscopy. In WT cells, the Mic60 clusters were seemingly distributed across the mitochondria, often resembling a stripe pattern perpendicular to the longitudinal axis of the mitochondria (Figs 4A and EV1A). In mitochondria of Mic10-KO cells, we found that Mic60 and Mic19 clusters localized in clearly discernible opposite distribution bands, i.e., they exhibited a two-sided distribution on the mitochondrial tubules (Figs 4A and EV2A). This opposite distribution of Mic60 clusters was also observed in Mic13-KO cells (Fig EV2B), showing that the absence of the Mic10-subcomplex influences the distribution of the Mic60-subcomplexes. Unexpectedly, also the $F_1F_o$-ATP synthase subunit b (ATPB) exhibited a similar two-sided distribution in Mic10-KO cells as visualized by 2D and 3D STED nanoscopy (Fig 4A and B). To further analyze the detailed localization of Mic60 in Mic10-KO cells, we visualized Mic60 with 3D MINFLUX nanoscopy (Gwosch *et al*, 2020). Thereby, we were able to localize immunolabeled Mic60 with an isotropic localization precision of about 5 nm in a ~ 300 nm thick volume. MINFLUX recordings confirmed the distribution of Mic60 clusters in two narrow opposite bands along the mitochondrion of a Mic10-KO cell (Fig 4C), which is entirely different to their stripe-like distribution in WT cells (Fig EV2C).

These observations raised the question, if the highly ordered distribution of the Mic60-subcomplex and of the $F_1F_o$-ATP synthase in Mic10-KO cells is just a consequence of the aberrant cristae morphology. Our ET data strongly suggested that the CMs form a hollow tube in the absence of Mic10. However, the ET data do not provide an ultimate proof for such a hollow tube, as the tomograms do not encompass a full mitochondrial tubule (Fig 3F). Therefore, we aimed to visualize the fold of the IM in the entire mitochondrial volume using live-cell fluorescence microscopy as well as FIB-SEM. We labeled the cells with Mito-tracker Green and imaged the IM using linear 3D structured illumination microscopy (3D SIM). 3D SIM of Mic10-KO cells fully supported the view that mitochondria devoid of Mic10 contained tube-like cristae that line the interior of the mitochondrion

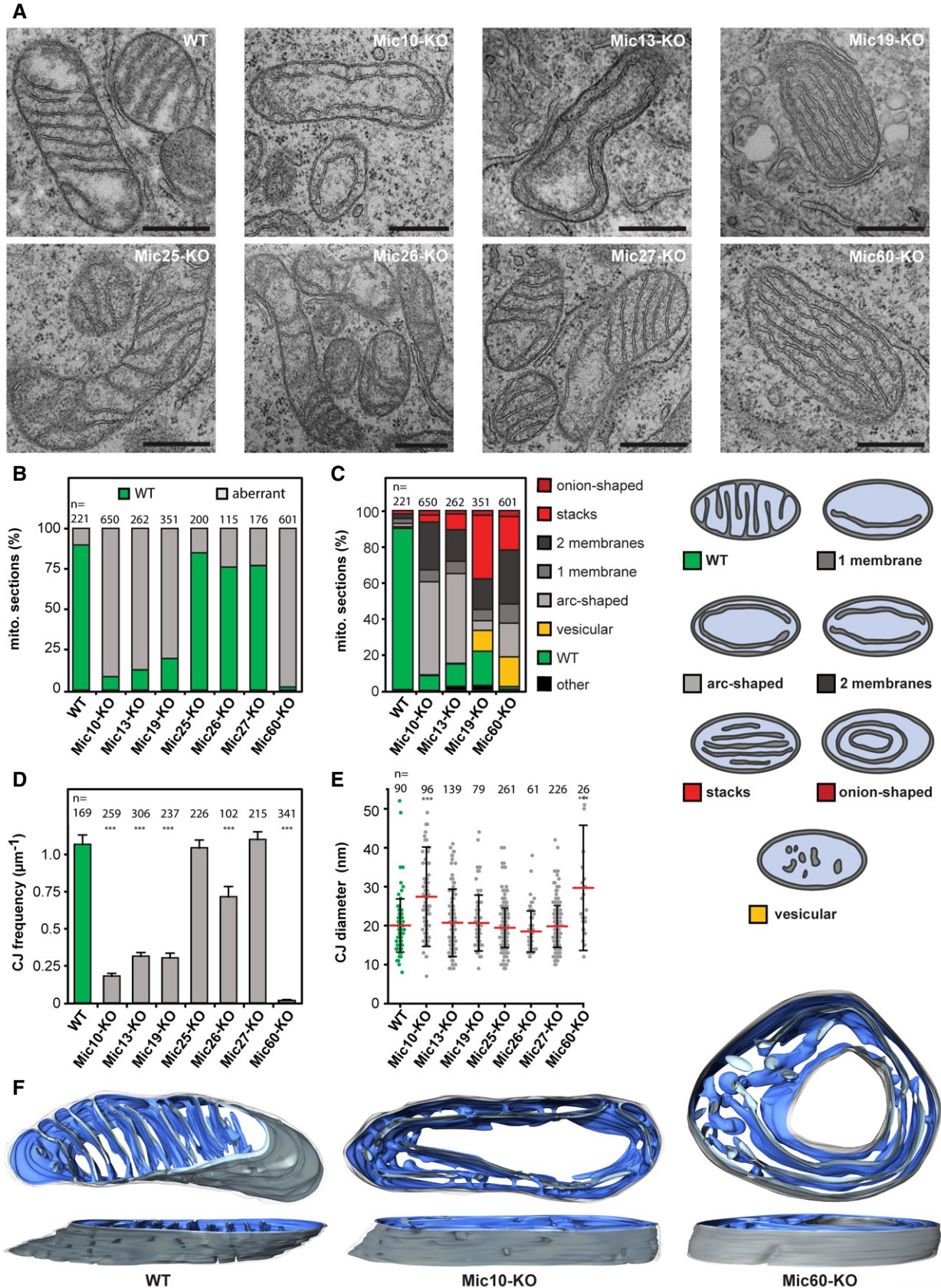

**Figure 3.**

**Figure 3.  Depletion of MICOS subunits affects the formation of lamellar cristae.**

A   Representative TEM recordings of WT cells and MICOS-KO mutants, as indicated.  Ultra-thin sections were taken in parallel to the growth surface of the cells.
B   Quantification of the overall cristae morphology on TEM recordings.
C   Quantification of detailed morphology of CMs in MICOS mutants (left). TEM recordings were assigned to eight classes (right).
D   CJ frequency on TEM recordings. The number of CJs was manually determined and normalized to the length of the OM. Samples were compared to WT by a one-way ANOVA test. Error bars: SEM.
E   Diameter of CJs. CJs were manually measured on TEM recordings. Red line: Average. Error bars: SD. Samples were compared to WT by a one-way ANOVA test.
F   Representative ET reconstructions of mitochondria from a WT cell (left), a Mic10-KO cell (center), and a Mic60-KO cell (right). The OM is displayed in clear gray; the side of the IM that faces the matrix is shown in dark blue. The IM side that faces the IMS is shown in light blue.

Data information: *n*: number of mitochondrial sections (B, C, E) or number of CJs (D). ***$P \leq 0.001$. Scale bars: 500 nm (A), 250 nm (F).

(Figs 4D and EV2D). Live-cell STED recordings of Mic10-KO cells expressing COX8A-SNAP further confirmed that the mitochondria in Mic10-KO cells contain mobile, large tube-like cristae (Movies EV8 and EV9). Finally, FIB-SEM of Mic10-KO cells unequivocally demonstrated the presence of tube-like cristae, which were perforated (Fig 4E). Because the edges of the perforations have a positive membrane curvature, we speculate that the $F_1F_O$-ATP synthases might decorate the perforations and therefore is localized at the rims of the mitochondria (Fig 4A and B).

The ET, FIB-SEM, and super-resolution data conclusively demonstrated that in Mic10-KO cells, the CMs form generally single-layered hollow tubes. It is difficult to reconcile the formation of the opposite Mic60 distribution bands with a large tube-like CM that evenly lines the mitochondrial tubule. We concluded that the distribution of the $F_1F_O$-ATP synthase and of the Mic60 clusters in opposite bands is not primarily determined by the shape of the tube-like CM. We assumed that the abundance of Mic10 controls the distribution of Mic60. To test this idea we analyzed the Mic60 distribution in Mic19-KO cells which exhibit reduced Mic60 levels and a strongly aberrant cristae architecture (Fig 3A and B), but remain residual levels of the Mic10 subcomplex (Fig 2A). In these cells, Mic60 was often found in randomly scattered clusters instead of opposite bands (Fig EV2E). In addition, we occasionally observed mitochondria that exhibited continuous ring- or arc-like Mic60 structures (Fig EV2E). The formation of such continuous Mic60 assemblies was strongly increased when we overexpressed Mic10-FLAG in Mic19-KO cells (Figs 4F and EV2F). In addition, Mic10-FLAG overexpression also raised the expression level of Mic60 in Mic19-KOs (Fig EV2F).

Altogether, these data demonstrate that the expression level of Mic10 influences the distribution of Mic60 and also of the $F_1F_O$-ATP synthase. In the absence of Mic10, Mic60 is found in clusters organized in opposite distribution bands, whereas at elevated Mic10 levels, Mic60 forms extended assemblies.

### Re-expression of Mic60 in Mic60-KO cells stabilizes MICOS and induces the formation of secondary CJs

To better understand the role of MICOS in cristae formation, we next analyzed how Mic60-deficient mitochondria respond to a rescue of the cellular Mic60 levels. To this end, we inserted the coding sequence of Mic60 under the transcriptional control of a tetracycline-/doxycycline-inducible (TetOn) promoter into the genome of Mic60-KO cells, thereby generating a Mic60-TO cell line.

After induction of the re-expression of Mic60 by adding doxycycline, we investigated the effects of increasing Mic60 levels over time (Fig 5). We titrated the concentration of doxycycline to reach approximately endogenous Mic60 expression levels after

48 h. After this time, virtually all cells expressed Mic60 (Fig 5A and B, Appendix Fig S3A). Mic60 re-expression rescued the cellular protein levels of all MICOS subunits (Fig 5B) and caused the re-formation of tubular mitochondrial networks (Fig 5C, Appendix Fig S3B).

Next, we asked whether the aberrant cristae in the Mic60-KO cells are converted to the WT morphology, or whether they are replaced by newly formed cristae. To address this question, we analyzed the morphology changes upon MICOS re-expression over time by analysis of numerous TEM recordings and classified the CM morphologies (Fig 5D and E). We observed a considerable number of mitochondria with intermediately shaped cristae; i.e., IM segments that partly showed a lamellar morphology and partly had an aberrant appearance (Fig 5D and E). These contiguous IM segments strongly point to a conversion of the aberrant cristae. Furthermore, we did not observe an increased number of small cristae that would hint to a strong increase of cristae biogenesis. With rising Mic60 levels, the number of CJs increased steadily over time and reached WT levels after about 48 h (Fig 5F). Importantly, we found that after 16 h of induction of Mic60 expression, only a fraction of the new CJs were connected to cristae with WT morphology (Fig 5G, Appendix Fig S4). Instead, a substantial part of the CJs induced by Mic60 expression was found on aberrant or intermediately shaped cristae. We conclude that these CJs are formed on already existing cristae and denote them  as secondary CJs.

### Formation of the holo-MICOS complex results in crista membrane remodeling and a redistribution of the CJs

The conversion of the cristae in the Mic60-TO cells is difficult to analyze because multiple layers of CM are involved. In comparison, the Mic10-KO has a less complex IM architecture, as the mitochondria usually exhibit only one layer of a tube-like CM that is already connected to the IBM by CJs (Fig 3A and F). Therefore, we next aimed to analyze the cristae conversion process upon re-expression of Mic10 in a Mic10-deficient cell line. To this end, we integrated a TetOn version of Mic10 C-terminally fused with a FLAG-T2A-EGFP tag into Mic10-KO cells. The self-cleaving T2A-peptide (Ryan *et al*, 1991) causes the release of cytosolic EGFP as an expression reporter, whereas the FLAG-tagged Mic10 is transported into the mitochondria (Figs 6A and EV3A). To test whether the re-expression of Mic10-FLAG induces the formation of the holo-MICOS complex, we performed Co-IPs from cell lysates using anti-FLAG and anti-Mic60 antibodies (Fig EV3B, Appendix Fig S5A). Using Mic10-FLAG as a bait, we pulled down Mic60, Mic26, Mic19, and Mic13 at levels mirroring the increasing Mic10 levels. When we

used Mic60 as a bait, we isolated Mic10, Mic13, and Mic26 at levels corresponding to the expression levels of Mic10, whereas Mic19 was isolated irrespective from the Mic10 expression levels (Fig EV3B). These findings demonstrate that re-expression of

Mic10-Flag rescues the expression levels of Mic13 and Mic26 and that newly synthesized subunits of the Mic10-subcomplex bind to the pre-existing Mic60-subcomplex to form the fully assembled MICOS complex.

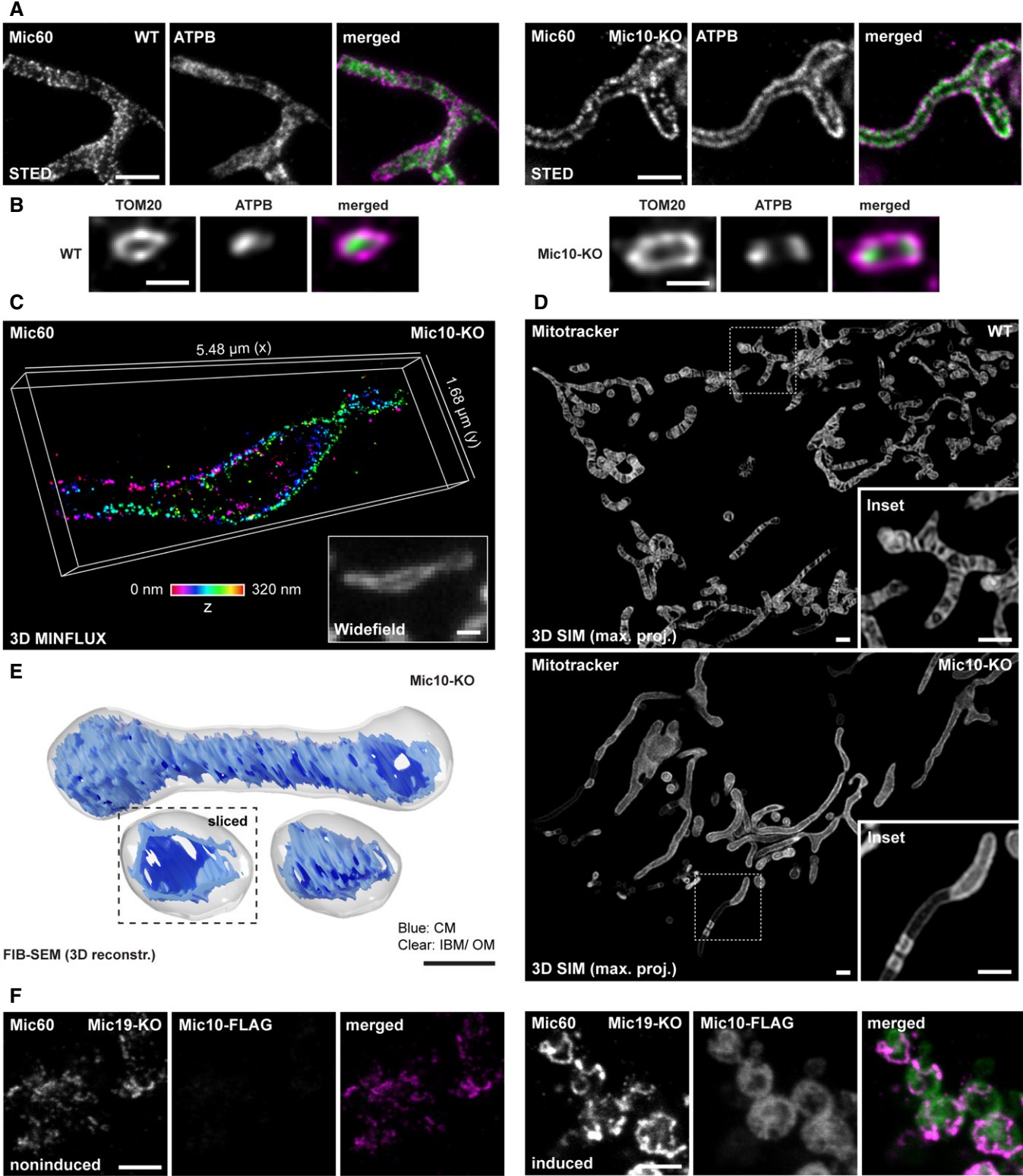

Figure 4.

◄

**Figure 4.   Depletion of Mic10 causes formation of large tube-like cristae and opposite distribution bands of Mic60 and ATPB.**

A, B  Nanoscopy of mitochondria of WT and Mic10-KO cells, as indicated. (A) Mitochondria were immunolabeled for Mic60 and ATPB and visualized by 2D STED nanoscopy. (B) Mitochondria were immunolabeled for TOM20 and ATPB and recorded with 3D STED nanoscopy. Image data were deconvolved. Depicted are cross sections of a mitochondrion (side view).

C     3D MINFLUX nanoscopy. Mic10-KO cells were immunolabeled for Mic60 using a directly labeled antibody. Colors encode depth information.

D     3D SIM of living WT and Mic10-KO cells. Cells were labeled with Mitotracker Green. Images show maximum intensity projections.

E     FIB-SEM of Mic10-KO cells. Shown is a representative reconstructed mitochondrion. Dashed box: Orthoslice view on the smaller mitochondrion. The OM and IBM are displayed together in clear gray, the CM in is shown in blue.

F     2D STED nanoscopy of mitochondria. Mic19-KO cells were transfected with a plasmid for Mic10-FLAG expression under the control of a tetracycline-inducible promoter. Left panel: Mitochondria from noninduced cell. Right: Mitochondria from induced cell expressing Mic10-FLAG.

Data information: Scale bars: 1 μm (A, C, D, F), 500 nm (B, E).

After 48 h of induction, the majority of cells expressed Mic10-FLAG and showed mitochondrial networks comparable to WT cells (Appendix Fig S5B and C). Moreover, the cristae morphology was rescued in the majority of these cells as demonstrated by live-cell STED recordings (Fig 6A and B). As the expression levels of Mic10 did not raise simultaneously in all cells, few cells still exhibited a low Mic10 expression level after 24 h of induction, whereas most cells already featured an elevated Mic10 expression level. In cells exhibiting low Mic10 expression levels, the faint Mic10-FLAG signals largely co-localized with Mic60 in narrow opposite distribution bands, comparable to the Mic60 distributions bands observed in Mic10-KO cells (Fig 6C). In accordance with the co-immunoprecipitation experiments (Fig EV3B), these observations suggest that Mic10-FLAG is recruited to the existing Mic60-subcomplexes. In cells showing higher Mic10-FLAG expression, Mic10 and Mic60 also largely co-localized, but their distribution in opposite bands was no longer obvious, as the protein clusters spread over the mitochondria (Fig 6C). Together, these data suggest that Mic10, after being recruited to the Mic60-subcomplex, changes the distribution of the holo-MICOS complex in the IM.

Next, we recorded 2D TEM images of mitochondria at different time points after re-expression of Mic10-FLAG in Mic10-TO cells and analyzed the progress of the structural rescue of the cristae (Fig 6D and E). After 24 h of induction, the cristae morphology, the number of CJs (Fig 6F), as well as the average CJ diameter (Fig 6G) resembled the situation in WT cells. The TEM recordings also revealed that the rescue of the crista phenotype seemed to involve an "intermediate" phenotype, i.e., contiguous cristae that were partly lamellar and partly tube-like (Fig 6D, inset, Appendix Fig S5D). These observations suggest a continuous remodeling of the aberrant tube-like cristae into lamellar cristae controlled by the assembly of the holo-MICOS complex. To test the idea of a continuous IM remodeling, we next analyzed the distribution of Mic10, Mic60, and ATPB in Mic10-TO cells induced for 16 h using STED nanoscopy (Fig EV3C–E). In line with the EM data, we observed intermediate phenotypes for the distribution of Mic10, Mic60, and ATPB, further supporting extensive IM remodeling induced by Mic10 re-expression.

To further investigate the reshaping of the cristae upon expression of Mic10-FLAG in 3D, we performed FIB-SEM that allows visualizing whole mitochondria in 3D, but lacks the resolution to discern individual CJs (Fig 7A, Appendix Fig S6). The FIB-SEM data verified the existence of intermediate cristae which are presumably developing from a tube-like structure into a lamellar shape (Fig 7A; Movies EV10–EV12). Representative reconstructions based on FIB-SEM data display cristae morphologies of a noninduced (0 h), an intermediate (16 h), and a largely rescued mitochondrion (24 h) from Mic10-TO cells (Fig 7A, Movies EV10–EV12). The 3D structure of the intermediate mitochondrion is suggestive of an arching of the large unfolded CMs (Fig 7A, lower panel, Movie EV11). We propose that these wavy CMs are subsequently converted into individual lamellar cristae.

Next, we analyzed Mic10-TO cells re-expressing Mic10 using ET. In contrast to FIB-SEM, the ET recordings allowed us to analyze the CJs, although the analyzed volume did not encompass an entire mitochondrial tubule. The representative reconstructions based on ET data demonstrate that the cristae in mitochondria of noninduced and rescued Mic10-TO cells are similar to the ones from Mic10-KO and WT cells, respectively (Fig 7B, Movies EV13 and EV14). Like Mic10-KO cells, noninduced cells showed irregularly distributed CJs that connected the tube-like cristae with the IBM (Fig 7B, Movie EV13). In induced cells, we observed mitochondria exhibiting separated lamellar cristae, representing the final stage of recovery, contiguous with interconnected, ragged cristae structures, corresponding to an intermediate stage (Fig 7B, Movie EV14). At lamellar cristae, the CJs were again arranged in line patterns perpendicular to the longitudinal axis of the mitochondrion (Fig 7B).

Taken together, we conclude that upon re-expression of Mic10-FLAG in Mic10-TO cells, the Mic10-subcomplex is stabilized and interacts with the Mic60-subcomplex. The formation of the holo-MICOS complex is accompanied by a redistribution of MICOS around the mitochondria and a recovery of the lamellar cristae. Our data suggest that during this process, the large tube-like cristae of the noninduced Mic10-TO mitochondria furrow, fragment, and are thereby converted into separated lamellar cristae.

**Outer membrane fission or fusion is not essential for lamellar cristae formation**

In yeast, the generation of lamellar cristae depends on the mitochondrial fusion machinery (Harner *et al*, 2016; Kojima *et al*, 2019). Because mitochondrial fission and fusion are balanced processes, fission defective mitochondria also exhibit a reduced number of fusion events. Indeed, mitochondria of Δ*dnm1* yeast cells, which have strongly reduced mitochondrial fission rates, exhibit a substantially reduced number of lamellar cristae, but a high number of branched, tubular cristae (Harner *et al*, 2016). In order to test if also in higher eukaryotes the formation of lamellar cristae is dependent on mitochondrial fission, we depleted WT cells of DRP1 (also known as DNM1L, dynamin-1-like protein),

which is essential for the fission of mitochondrial tubules in higher eukaryotes (Bleazard *et al*, 1999; Lee *et al*, 2004; Cereghetti *et al*, 2008). Corroborating previous reports (Ban-Ishihara *et al*, 2013), WT cells depleted of DRP1 by RNAi for 7 days exhibited hyperfused mitochondrial networks and numerous lamellar cristae (Fig EV4A and B). This observation suggests that

in HeLa cells, fission of mitochondrial tubules is not essential for the generation of lamellar cristae.

In mammalian cells, the fusion of the mitochondrial OM is regulated by the two mitofusins MFN1 and MFN2, two highly conserved dynamin-related GTPases, which exhibit distinguishable functions (Ishihara *et al*, 2004; Giacomello *et al*, 2020). To investigate

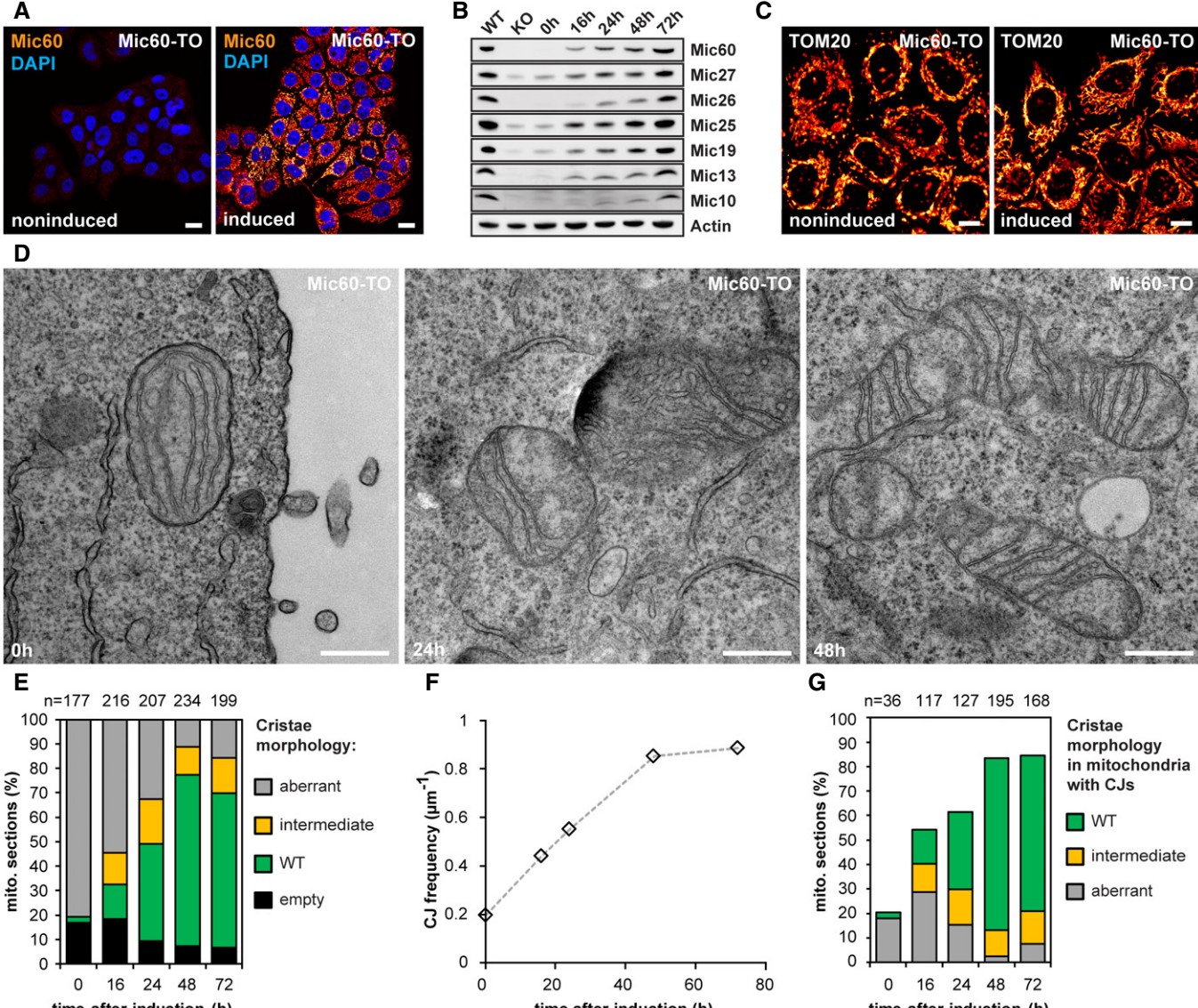

**Figure 5. Mic60 controls MICOS protein levels, CJ frequency, and inner membrane morphology.**

Rescue of Mic60 expression in Mic60-TO cells upon induction with doxycycline hyclate.

A   Cells were immunolabeled for Mic60 and DNA and visualized by confocal fluorescence microscopy before and after induction of Mic60 expression for 48 h.

B   Protein levels of MICOS proteins after Mic60 induction. Cell lysates were analyzed by Western blotting at the indicated time points.

C   Recovery of mitochondrial networks upon Mic60 re-expression. Cells were induced with doxycycline hyclate for 72 h, immunolabeled for TOM20, and visualized by confocal microscopy.

D–G   TEM of Mic60-TO cells before and after induction. (D) TEM recordings of Mic60-TO cells at the indicated time points. (E) Quantification of the cristae morphology. (F) CJ frequency. The number of CJs on TEM recordings was normalized to the length of the OM. At least 90 mitochondrial sections were analyzed for each sample. (G) Quantification of the cristae morphology of Mic60-TO cells. Only mitochondria exhibiting at least one CJ were analyzed.

Data information: *n*: Number of mitochondrial sections. Scale bars: 20 μm (A), 10 μm (C), 500 nm (D).
Source data are available online for this figure.

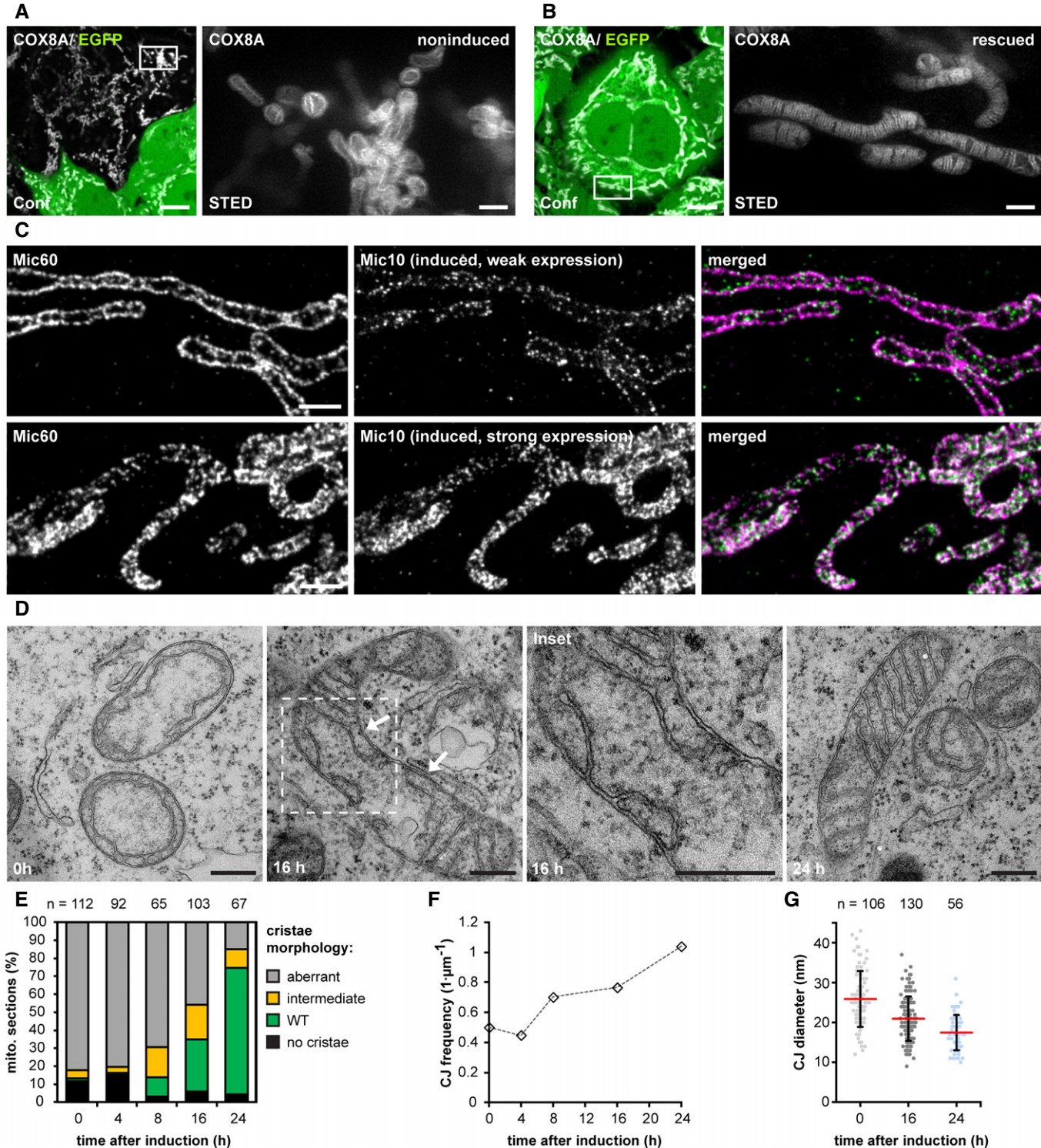

**Figure 6.  The Mic10-subcomplex is essential for coordination of lamellar cristae formation.**

A, B   Live-cell nanoscopy of Mic10-TO cells. Cells expressing COX8A-SNAP were labeled with SNAP-cell SiR and recorded with STED nanoscopy. (A) Cristae architecture in noninduced cells. (B) Cristae architecture after Mic10 re-expression (24 h induction with doxycycline). STED image data were deconvolved.

C   STED nanoscopy of fixed Mic10-TO cells. Cells were induced for 24 h and immunolabeled for Mic60 and Mic10-FLAG. Upper row: Cell with weak Mic10 expression level. Lower row: Cell with strong Mic10 expression level.

D–G   TEM of Mic10-TO cells. (D) Cells induced for the indicated period of time were recorded with TEM. Arrow indicates an intermediately shaped crista. (E) Quantification of the cristae morphology. *n*: Number of mitochondrial sections. (F) CJ frequency in Mic10-TO cells. Number of CJs was normalized to the length of the OM. The same sections were analyzed as in (E). (G) Diameter of CJs estimated on TEM recordings.

Data information: *n*: number of analyzed mitochondrial sections (E) or CJs (G). Error bars: SD. Scale bars: 10 µm (A, B, conf), 1 µm (A, B, C), 0.5 µm (D).

whether OM fusion is essential for lamellar crista formation, we depleted HeLa cells of MFN1 or MFN 2 or MFN1 together with MFN2. Depletion of these proteins resulted in a mild cristae phenotype, but lamellar cristae were still observed (Fig EV4C and D).

We conclude that in mammalian cells OM fission or fusion is not essential for the development of lamellar cristae.

## Deletion of OPA1 induces a moderate cristae phenotype in WT cells

A substantial body of evidence demonstrates that the dynamin-like GTPase OPA1 (Mgm1 in yeast) influences cristae architecture (Ramonet *et al*, 2013; Patten *et al*, 2014; Cogliati *et al*, 2016;

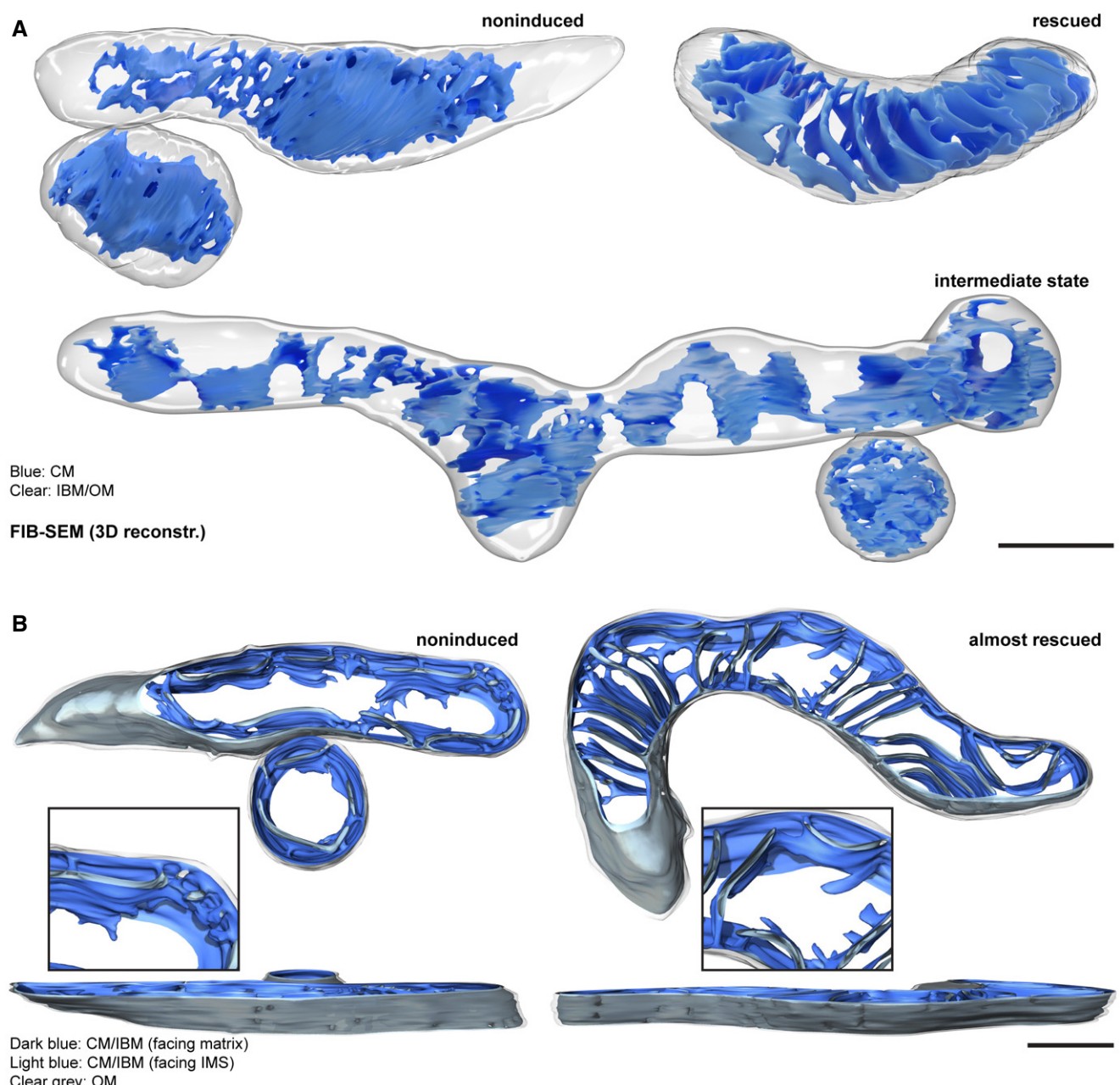

**Figure 7. holo-MICOS complex assembly induces remodeling of the mitochondrial inner membrane.**

A FIB-SEM of Mic10-TO cells. Cells were induced for 0, 16, and 24 h. The representative reconstructions show mitochondria from noninduced, rescued, and intermediate cells, as indicated. The OM is shown together with the IBM in clear gray, the CM in blue.

B ET of Mic10-TO cells. Mic10 expression was induced for 0 and 16 h. Shown are reconstructions of representative noninduced and almost completely rescued mitochondria. The OM is displayed in clear gray; the side of the IM that faces the matrix is shown in dark blue. The IM side that faces the IMS is shown in light blue.

Data information: Scale bars: 500 nm.

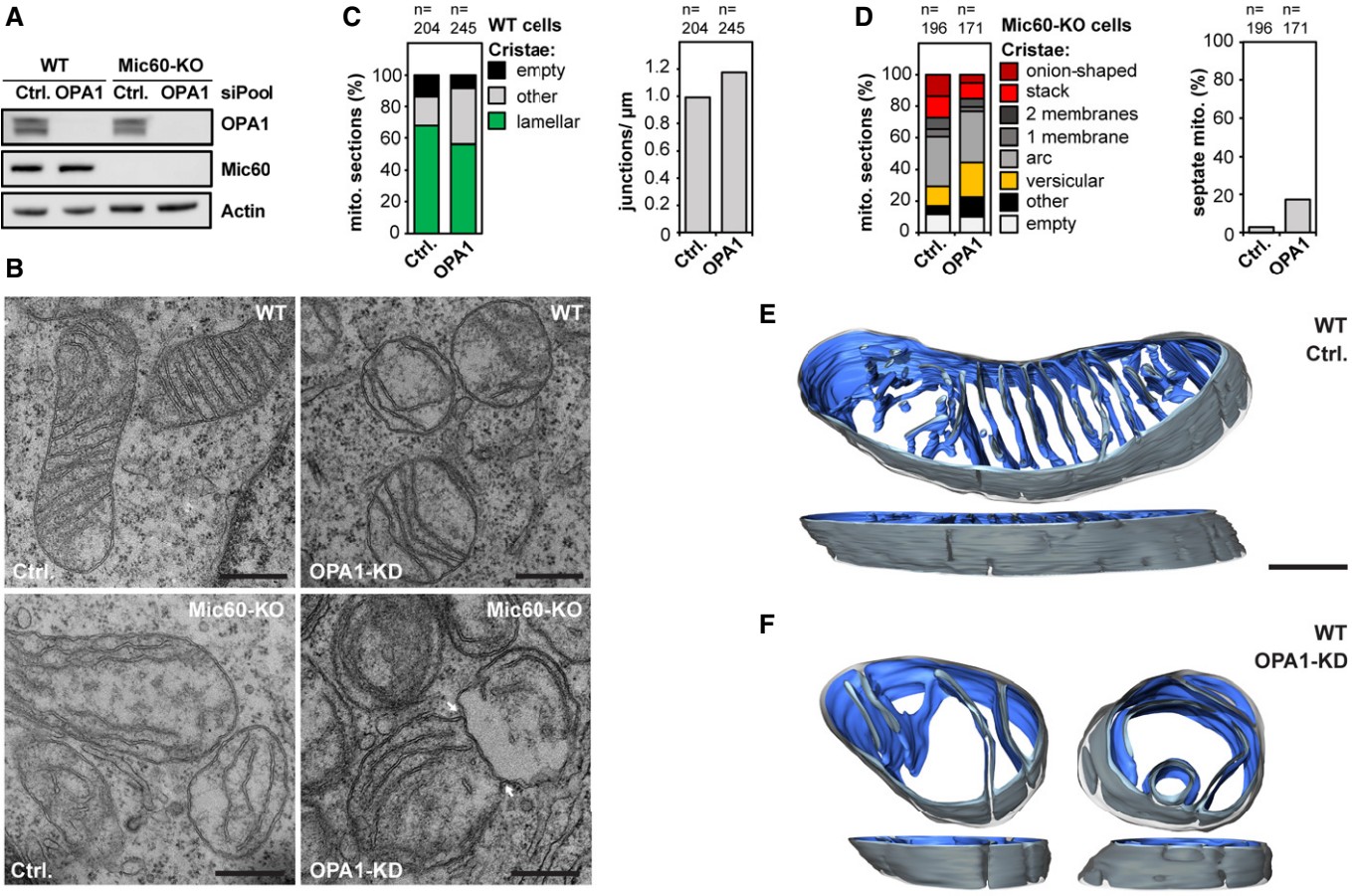

**Figure 8. OPA1 influences the inner membrane morphology.**

Knockdown (KD) of OPA1 in HeLa cells. WT and Mic60-KO cells were transfected with a scrambled control (Ctrl.) or siRNA pools against OPA1 (OPA1-KD) for 5 days.

A     Cell lysates were analyzed by Western blotting.
B     TEM recordings from OPA1-depleted WT and Mic60-KO cells. Arrows point to a septum.
C, D   Quantification of the cristae morphology of OPA1 deficient WT (C) and Mic60-KO (D) cells. The number of CJs was normalized to the length of the OM (C). The number of septate mitochondria in Mic60-KO cells was estimated (D).
E, F   ET of mitochondria from OPA1-KD cells. WT cells were transfected with a scrambled control (E) or siRNA pools against OPA1 (F) for 48 h. Reconstructions of representative mitochondria are displayed. The OM is displayed in clear gray; the side of the IM that faces the matrix is shown in dark blue. The IM side that faces the IMS is shown in light blue.

Data information: Scale bars: 500 nm (B), 250 nm (E, F).
Source data are available online for this figure.

Glytsou *et al*, 2016; MacVicar & Langer, 2016; Kondadi *et al*, 2019). To further investigate the role of OPA1 in cristae development, we first efficiently depleted OPA1 by RNAi in WT and Mic60-KO cells (Fig 8A). Compared to Mic10-KO and Mic60-KO cells, OPA1-depleted WT cells had a moderate cristae phenotype, as the portion of mitochondrial TEM sections showing cristae lamellae was reduced by only about 10%; still the cristae often appeared shorter, disordered, or partly swollen (Fig 8B and C), fully in line with previous studies (Olichon *et al*, 2003; Barrera *et al*, 2016; Glytsou *et al*, 2016).

Yeast Δ*mgm1* cells have been reported to contain septa, i.e., IM structures that divide the mitochondrial matrix in two physically separated compartments (Sesaki *et al*, 2003; Harner *et al*, 2016; Kojima *et al*, 2019). Such septa are the result of a lack of

IM fusion after tubule fusion (Harner *et al*, 2016). In WT HeLa cells, a single crista is often connected to the IBM by CJs on both sides of the mitochondrion (Fig 3A). Thus, septa could be mistaken by cristae on TEM recordings. However, in Mic60-KO cells, septa junctions should be unequivocally recognized as almost no CJs are formed in the absence of Mic60. Indeed, we observed in ~ 15% of the mitochondria from OPA1-deficient Mic60-KO cells such septa (Fig 8B and D). Otherwise, the mitochondria were phenotypically similar to the scrambled RNAi Mic60-KO control (Fig 8B and D). The observation of septa in OPA1-depleted Mic60-KO cells suggested that such septa also exist in OPA1 depleted WT cells. Therefore, a fraction of CJs in these cells (Fig 8C) might actually represent septa junctions. To determine if in WT cells depleted of OPA1 the majority of the

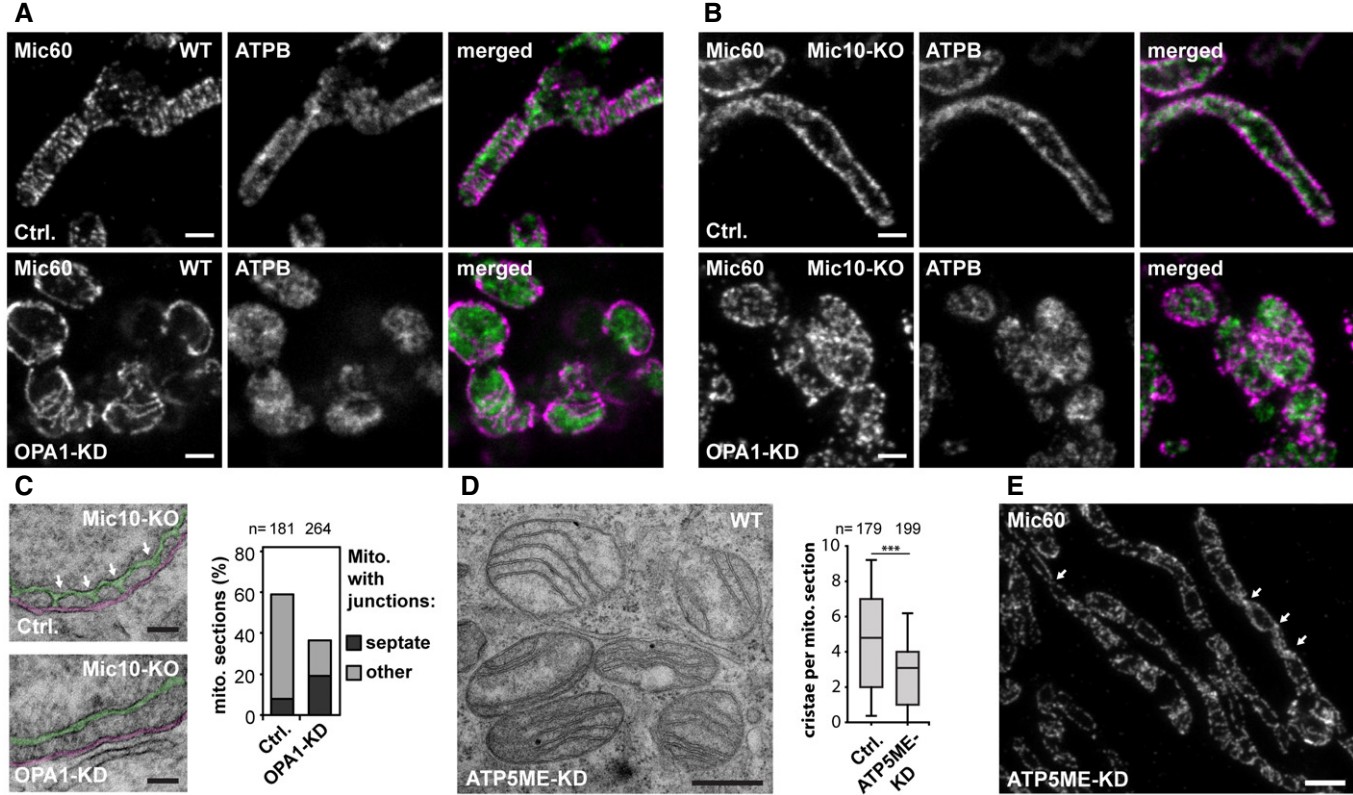

**Figure 9. OPA1 affects the Mic60 distribution and stabilizes tubular CJs.**

A–C WT and Mic10-KO cells were transfected with a scrambled control (Ctrl.) or siRNA pools against OPA1 (OPA1-KD) for 72 h. (A, B) STED nanoscopy of OPA1-depleted WT (A) and Mic10-KO (B) cells. Cells were immunolabeled for Mic60 and ATPB. (C) TEM of OPA1-depleted Mic10-KO cells. Left: Representative TEM recordings showing CJs marked by arrows in a control cell and a similar view for an OPA1-KD cell. The CMs are colored in green and the OM and IBM together in magenta. Right: Quantification of mitochondria containing CJs or septa junctions. Sections that contained CJ-like structures were analyzed. Of these, the number of sections exhibiting septa or cristae structures (other) was quantified.

D, E Depletion of ATP5ME in HeLa cells. Cells were transfected with a scrambled control or siRNA pools against ATP5ME for 4 days. (D) TEM recording (left) and quantification of the number of cristae per mitochondrial section (right). (E) Cells depleted of ATP5ME were immunolabeled for Mic60 and recorded with STED. The arrows point to crossing points of the opposite Mic60 distribution bands.

Data information: $n$: Number of mitochondrial sections analyzed. Boxes indicate 25th to 75th percentile. Horizontal lines indicate median. Whiskers indicate SD. Mann–Whitney test was used to compare samples. ***$P \leq 0.001$. Scale bars: 500 nm (A, B, D), 100 nm (C), 1 µm (E).

IM structures are septa or cristae, we performed ET (Fig 8E and F, Movies EV15 and EV16). Reconstructions revealed that most of the IM structures did not continuously cross the entire mitochondrion within the ~ 200 nm thick sections and therefore represent disordered lamellar cristae, not septa (Fig 8F, Movie EV16).

## OPA1 influences the formation of MICOS assemblies and stabilizes tubular CJs

### OPA1 and Mic10 antagonistically influence the size and the distribution of Mic60 assemblies

We next investigated the influence of OPA1 on the distribution of Mic60. To this end, we immunolabeled cells for Mic60 and the ATPB subunit of the $F_1F_o$-ATP synthase and performed STED nanoscopy (Fig 9A). In OPA1-deficient mitochondria, we found conspicuous ring- and rib-like Mic60 assemblies that were significantly larger than the rod-shaped Mic60 clusters in the WT control

(Fig 9A). Because in Mic19-deficient cells the occurrence of elongated Mic60 assemblies was Mic10-dependent (Figs 4F and EV2F), we next asked if Mic10 also controls the formation of the Mic60 assemblies in OPA1-depleted cells. We found that in Mic10-KO cells devoid of OPA1, Mic60 did not form extended assemblies but localized in small clusters (Figs 9B and EV4E). In these cells, also the distribution of Mic60 in opposite distribution bands had disappeared, as the clusters were scattered across the mitochondria (Fig 9B). To test if the formation of large Mic60 assemblies in cells depleted of Mic10 and OPA1 can be induced by re-expression of Mic10, we depleted Mic10-TO cells of OPA1 for 48 h and subsequently induced Mic10 expression for 24 h (Fig EV4F). STED images showed that Mic10-FLAG expression resulted in the redistribution of Mic60 clusters into larger Mic60 assemblies (Fig EV4F). In addition, TEM demonstrated that OPA1-depleted Mic10-TO cells expressing Mic10-FLAG for 24 h developed disordered cristae lamellae, comparable to OPA1-depleted WT cells (Fig EV4G). This demonstrates that Mic10 induces cristae remodeling also in OPA1-

deficient cells. We conclude that OPA1 is not essential for the formation of cristae lamellae but controls, together with Mic10, the formation and distribution of Mic60 assemblies.

### OPA1 stabilizes tubular CJs in Mic10-KO cells

Since the simultaneous depletion of both Mic10 and OPA1 led to scattered Mic60 clusters, we next investigated if these cells still contained CJs. TEM of OPA1-depleted Mic10-KO cells revealed, except for a strong fragmentation of the mitochondrial tubules, a similar cristae phenotype as seen in Mic10-KO cells (Fig EV4H). However, we observed a reduction of CJs in Mic10-KO cells devoid of OPA1 (Fig 9C). In Mic10-KO cells transfected with scrambled siRNAs about 60% of the TEM sections of mitochondria showed at least one CJ, whereas after depletion of OPA1, only 36% of the mitochondria showed at least one CJ (Fig 9C). About 50% (about 15% in controls) of these junctions were associated with septa and thus are likely to represent septa junctions (Figs 9C and EV4I). Hence, in non-septate mitochondria devoid of Mic10 and OPA1, the number of CJs was decreased by 66%. These observations suggest an important role of OPA1 for the stability of tubular CJs in the absence of Mic10.

Taken together, OPA1 is required for proper cristae architecture, although its depletion results only in a mild phenotype compared to Mic10 or Mic60 knockouts. A simultaneous depletion of both OPA1 and Mic10 further reduced the number of CJs suggesting that OPA1 works in concert with the Mic10-subcomplex to stabilize tubular CJs. Additionally, OPA1 and Mic10 antagonistically determine the distribution and size of Mic60 assemblies.

### The dimeric $F_1F_o$-ATP synthase affects the cristae architecture and the Mic60 distribution

Next to MICOS and OPA1, the $F_1F_o$-ATP synthase is a key player for shaping the IM (Kuhlbrandt, 2019). Mitochondrial $F_1F_o$-ATP synthases generally form dimers, which can assemble into long ribbons that contribute to the shape of the cristae (Dudkina *et al*, 2006; Strauss *et al*, 2008; Blum *et al*, 2019). The $F_1F_o$-ATP synthase subunit ATP5ME (also known as ATP synthase subunit e, ATP5K or ATP5I) is involved in $F_1F_o$-ATP synthase dimerization (Arnold *et al*, 1998; Habersetzer *et al*, 2013; Quintana-Cabrera *et al*, 2018). To disturb dimer formation, we used RNAi to deplete ATP5ME in HeLa cells (Fig EV5A). Cells devoid of ATP5ME contained mostly large spherical mitochondria, but also some elongated tubular mitochondria (Figs 9D and EV5A and B). Compared to the depletion of MICOS or of the Mic10-subcomplex, the IM generally exhibited a mild phenotype. As reported previously, the mitochondria of cells depleted of ATP5ME contained fewer, often slightly disordered lamellar cristae (Figs 9D and EV5B) (Arnold *et al*, 1998; Paumard *et al*, 2002; Rabl *et al*, 2009; Habersetzer *et al*, 2013; Quintana-Cabrera *et al*, 2018).

As the $F_1F_o$-ATP synthase has been shown to interact with the MICOS complex in yeast (Eydt *et al*, 2017; Rampelt & van der Laan, 2017; Rampelt *et al*, 2017a), we next investigated the distribution of Mic60 in cells depleted of ATP5ME. In the spherical, more aberrant mitochondria, Mic60 seemed to be arranged in long stripes encircling the organelles (Fig EV5B). In tubular mitochondria, Mic60 primarily formed clusters that were localized on opposite sides of the mitochondrial tubules (Fig 9E). Between these bands, we

occasionally observed a stripe-like arrangement of Mic60 clusters perpendicular to the tube longitudinal axis (Fig 9E). This overall distribution of Mic60 in ATP5ME-depleted cells (Fig 9E) was reminiscent of the Mic60 distribution in Mic10-KO cells (Fig 4A), or in cell types that exhibit fewer or smaller cristae (Stoldt *et al*, 2019). Similar to the situation in these cell types, the narrow Mic60 bands were also often twisted, resulting in a helical arrangement of the Mic60 clusters (Fig 9E). The ATP5ME-dependent re-localization of MICOS suggests that the $F_1F_o$-ATP synthase dimers support an even distribution of MICOS, and consequently of the CJs, around the mitochondrial tubules. Indeed, when using Mic10-FLAG as a bait in co-IP experiments, we co-isolated also ATP5B, suggesting a physical interaction of the $F_1F_o$-ATP synthase with the Mic10 subcomplex in human cells (Appendix Fig S5A).

Taken all together, we conclude that in human cells, the two MICOS subcomplexes have different functions. The Mic60-subcomplex, which is stable in the absence of the Mic10-subcomplex, is essential for the maintenance of CJs and the stability of the holo-MICOS complex. The Mic10-subcomplex is essential for lamellar cristae formation. Formation of the holo-MICOS complex mediates extensive remodeling of pre-existing unstructured cristae into individual lamellae and also the formation of secondary CJs. We found that both OPA1 and Mic10 differently influence the distribution and size of MICOS assemblies. Finally, our data show that dimers of the $F_1F_o$-ATP synthase influence the cristae shape as well as the distribution of the MICOS complex, together demonstrating multiple functional interactions of the membrane-shaping proteins involved in cristae formation. Thereby, as detailed below, our findings support a new model for the formation of cristae in higher eukaryotes.

## Discussion

In this work, we investigated the interplay of the major determinants of cristae formation in higher eukaryotes, namely the Mic10 and the Mic60 subcomplexes of MICOS, OPA1, and the $F_1F_o$-ATP synthase. Our data suggest significant differences in cristae formation of higher and lower eukaryotes. We demonstrate that cristae development in human mitochondria is largely independent from mitochondrial fusion–fission dynamics, whereas fusion of mitochondria is essential for lamellar cristae formation in the yeast *Saccharomyces cerevisiae* (Harner *et al*, 2016). Another significant difference exists in the stability of the Mic10-subcomplex that depends on the Mic60-subcomplex in human cells, whereas it assembles in a Mic60-independent manner in yeast (von der Malsburg *et al*, 2011; Bohnert *et al*, 2015; Friedman *et al*, 2015; Guarani *et al*, 2015; Anand *et al*, 2016).

This study shows that the aberrant IM structures of Mic10-KO and Mic60-KO cells are converted into WT cristae upon re-formation of MICOS, rather than being replaced by new normally shaped CMs. The observed repair mechanisms allow us to draw conclusions on the mechanisms that are involved in *de novo* cristae biogenesis (Fig 10A). Furthermore, the fact that human Mic10-KO cells still form CJs, but exhibit an aberrant cristae architecture, allowed us to disentangle CJ formation from lamellar cristae formation and to investigate the distinct functions of the two MICOS subcomplexes.

### Opposite distribution bands

Our STED and 3D MINFLUX data show that in mitochondria of Mic10-KO cells, the Mic60 clusters are distributed along two narrow opposite distribution bands. As our FIB-SEM, ET, and 3D SIM data consistently show that in the absence of the Mic10-subcomplex, the cristae are large rotationally symmetric tube-like structures that line the IBM, the distribution of Mic60 in

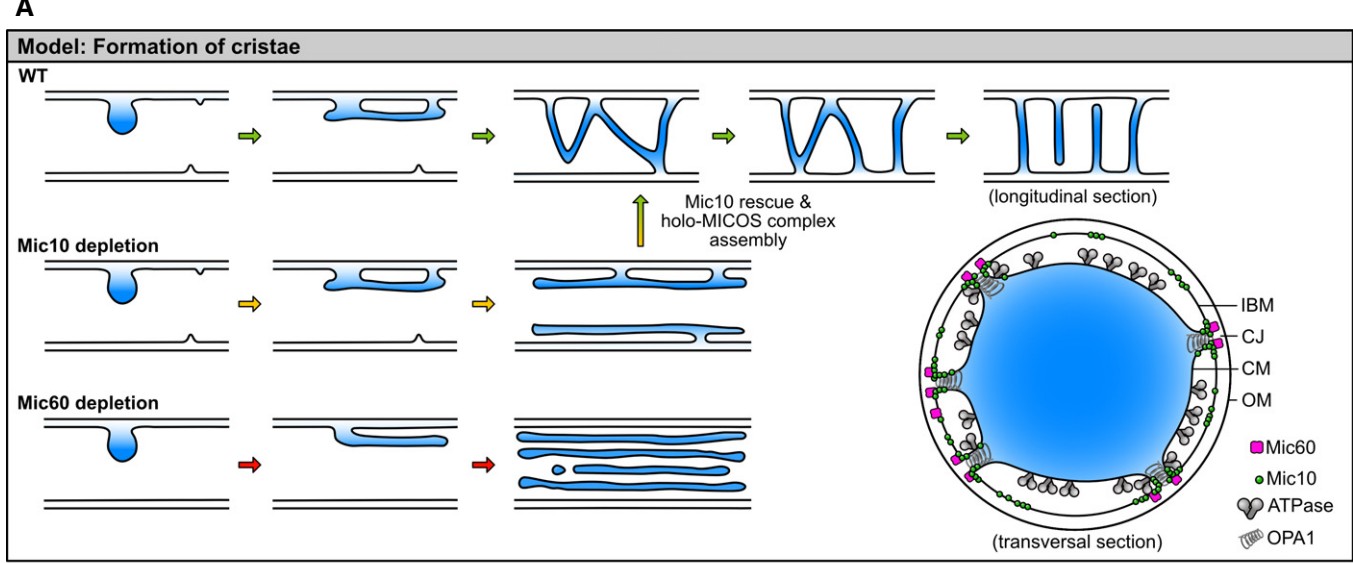

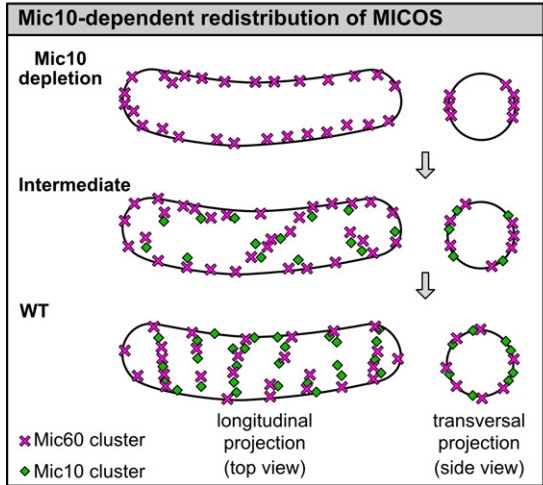

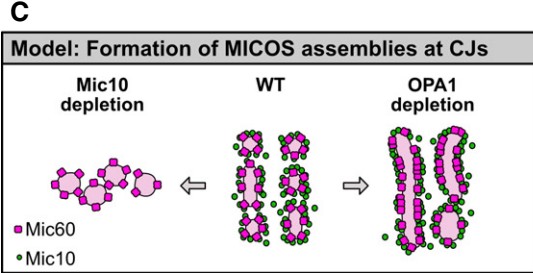

| | cristae shape | Mic60 distribution | crista junctions |
|---|---|---|---|
| **WT** | • lamellar | • clusters or small rods<br>• often arranged in transverse stripes | • circular or short slit-like CJs<br>• often arranged in transverse stripes |
| **Mic10 depletion** | • tube-like | • clusters<br>• arranged in two opposite bands | • number reduced<br>• circular shape<br>• diameter enlarged |
| **Mic60 depletion** | • tube-like<br>• onion-shaped<br>• stacks | • no Mic60 present | • number strongly reduced |
| **OPA1 depletion** | • lamellar (disordered)<br>• septa | • extended Mic60 assemblied | • slit-like CJs<br>• in addition septa junctions |
| **OPA1 & Mic10 depletion** | • tube-like<br>• vesicular<br>• septa | • clusters<br>• scattered distribution | • number strongly reduced<br>• in addition septa junctions |
| **OPA1 & Mic60 depletion** | • tube-like<br>• onion-shaped<br>• stacks<br>• septa | • no Mic60 present | • number strongly reduced<br>• in addition septa junctions |

**Figure 10. Summary of findings and model of MICOS-controlled lamellar crista formation.**

A Model for the formation of crista membranes (CMs) in WT, Mic10-KO, and Mic60-KO cells. Shown are cartoons of longitudinal cross sections of mitochondria. For details, see main text. Right lower corner: Model for the localizations of the key membrane-shaping proteins involved in lamellar cristae formation at a lamellar crista in WT cells. Shown is a transversal cross section through a mitochondrial tubule (view on a single crista). The CM is displayed in blue.

B Illustration of the Mic60 redistribution upon re-expression of Mic10 in Mic10-depleted mitochondria.

C Model of the Mic10- and OPA1-dependent formation of MICOS assemblies at CJs.

D Table summarizing the phenotypes that were observed in this study upon the depletion of key players in cristae formation.

opposite distribution bands is presumably not a consequence of the cristae morphology. In fact, such Mic60-distribution bands, which can adopt different width, have been previously reported in several WT cell types (Jans *et al*, 2013; Stoldt *et al*, 2019). However, this enrichment of Mic60 clusters on opposite sides of the mitochondrial tubules was almost invisible in the WT cells we used in this study. Different to the situation in, e.g., yeast cells or primary fibroblasts, these WT HeLa cells exhibit large well-developed lamellar cristae, each featuring several adjacent CJs, which almost encircle a mitochondrion. Therefore, in these cells the visibility of the Mic60 distribution bands, which reflect the CJ distribution, might be concealed because they are much wider or highly twisted. We show in this study that the redistribution of Mic60 in such opposite distribution bands can be induced by several means, including the depletion of ATP5ME, of Mic13, or of Mic10. The prevalence of this phenomenon suggests that these opposite distribution bands are an important structure providing element of mitochondria (Jans *et al*, 2013; Stoldt *et al*, 2019).

### Distribution of CJs

This study also demonstrates that the Mic60-subcomplex is necessary for the formation of CJs, whereas the Mic10 subcomplex is required for lamellar cristae formation. The remodeling of the IM and the generation of secondary CJs during re-expression of Mic10 in Mic10-TO cells are accompanied by a redistribution of MICOS and of the CJs (Figs 6C, 10B and EV3C–E). Next to the Mic10-subcomplex, the dimeric $F_1F_o$-ATP synthase, as well as OPA1, influences the distribution of the Mic60-subcomplex. Specifically, in the absence of OPA1, Mic10 induces the formation of extended Mic60 structures, suggesting that OPA1 restricts the size of MICOS assemblies. These findings point to antagonistic functions of OPA1 and Mic10 in the regulation of the distribution and size of the Mic60 subcomplexes (Fig 10C and D).

### Recovery of the Mic10 phenotype provides hints for cristae biogenesis

Remarkably, in yeast the detached IM structures of MICOS-deficient cells feature a similar protein composition as CMs in WT cells (Harner *et al*, 2014, 2016), suggesting that not CM formation, but the shaping of the IM is defective in MICOS-deficient cells. Upon re-expression of Mic10 in Mic10-TO cells, unstructured large CMs form an undulating pattern and develop into multiple lamellar cristae (Figs 7A and 10A). Concurrently, MICOS assembly triggers a redistribution of the CJs and the formation of new, secondary CJs, as observed in Mic10-TO and Mic60-TO cells (Figs 5F and G, 6C, and 10A and B). Therefore, the assembly of the holo-MICOS complex can be regarded as a switch controlling the efficient conversion of unstructured cristae into lamellar cristae (Fig 10A). We postulate that the unstructured cristae found in Mic10-KO cells represent a trapped intermediate structure that also occurs during normal cristae formation. We propose that in WT mitochondria, these intermediates are smaller and short lived and are rapidly re-shaped into lamellar cristae following the same principle as observed in rescued Mic10-TO cells (Figs 6D, 7A, 10A and B).

### A new model of cristae biogenesis

Our findings suggest that lamellar cristae biogenesis in higher eukaryotes starts in a MICOS independent way by an unstructured infolding of the IM, followed by MICOS-controlled restructuring of this infolding, including secondary CJ formation (Fig 10A). The positioning and shape of the CJs are fine-tuned by an interplay between the Mic10-subcomplex, OPA1, and the $F_1F_o$-ATP synthase. As lamellar cristae usually occur in densely stacked groups (Stephan *et al*, 2019), it is tempting to assume that the cristae of one group originate from a single precursor CM (Fig 10A). In mitochondria with a different cristae architecture, this cristae biogenesis pathway might also prove to be prevalent, as the reshaping of a larger cristae precursor into several individual cristae could principally lead to any cristae shape.

## Materials and Methods

### Materials availability

Further information and requests for resources and reagents should be directed to the corresponding author, Stefan Jakobs (sjakobs@gwdg.de).

### Cell culture and transfection

HeLa cells (Gruber *et al*, 2005) were grown in Dulbecco's modified Eagle's medium (DMEM) with glutaMAX and 4.5 g/l glucose (Thermo Fisher Scientific, Waltham, MA, USA) supplemented with 100 U/ml penicillin and 100 μg/ml streptomycin (Merk Millipore, Burlington, MA, USA), 1 mM sodium pyruvate (Sigma-Aldrich, Munich, Germany), and 10% (v/v) fetal bovine serum (Merck Millipore) at 37°C and 5% $CO_2$. For gene silencing by RNA interference (RNAi), cells were transfected with siRNA pools (siTOOLs Biotech, Planegg, Germany) according to the manufacturer's instruction using the Lipofectamine RNAiMAX transfection reagent (Thermo Fisher Scientific). Plasmid transfections were carried out using jetPRIME (Polyplus-transfection SA, Illkirch-Graffenstaden, France) or FuGENE HD (Promega, Fitchburg, WI, USA).

### Generation of knockout cell lines by CRISPR/Cas9

Sequence information about each target gene was collected from the gene database of the National Center for Biotechnology Information (NCBI). Each gRNA was designed using the CRISPR design tool from Benchling based on the scoring models from (Hsu *et al*, 2013; Doench *et al*, 2016). For the cloning of the nuclease plasmids, the expression vector PX458 was digested with the BbSI restriction endonuclease (New England Biolabs, Ipswich, MA, USA) and purified. Oligonucleotides were hybridized and ligated into PX458. pSpCas9(BB)-2A-GFP (PX458) was a gift from Feng Zhang (Addgene plasmid #48138; http://n2t.net/addgene:48138; RRID: Addgene_48138). HeLa cells were transfected with the respective nuclease plasmid and cells expressing Cas9-EGFP were sorted using a BD Influx cell sorter (BD Biosciences, Flanklin Lakes, NJ, USA) 4 days after transfection. After clonal expansion, the single-cell clones were analyzed by SDS–PAGE and Western blotting. Gene

disruption was verified by PCR of the target gene, sub-cloning, and sequencing. For primers, see Table 1.

## Generation of stable Mic10 and Mic60 TetOn (TO) cell lines by CRISPR/Cas9

### Cloning of donor plasmids

#### pTRE-Tight-Mic60

The plasmid pTRE-Tight-EGFP-donor fw copy was linearized by PCR. Mic60 was amplified by PCR and both fragments were ligated by Gibson Assembly (New England Biolabs). pTRE-Tight-EGFP-donor fw copy was a gift from Rudolf Jaenisch (Addgene plasmid # 22074; http://n2t.net/addgene:22074; RRID:Addgene_22074).

#### AAVS1-TRE3G-Mic10-FLAG-T2A-EGFP

The plasmid AAVS1-TRE3G-EGFP was linearized by the restriction endonuclease SalI. Mic10-FLAG was amplified by PCR, thereby also introducing a C-terminal T2A self-cleavage site. Fragments were ligated by Gibson Assembly (New England Biolabs). AAVS1-TRE3G-EGFP was a gift from Su-Chun Zhang (Addgene plasmid #52343; http://n2t.net/addgene:52343; RRID:Addgene_52343).

#### PX-330-AAVS1

The nuclease plasmid PX330-AAVS1 was derived from PX330. In brief, oligonucleotides were annealed by primer annealing and integrated into PX330 after linearization with the BbSI restriction endonuclease. pX330-U6-Chimeric_BB-CBh-hSpCas9 was a gift from Feng Zhang (Addgene plasmid #42230; http://n2t.net/addgene: 42230; RRID:Addgene_42230). For primers, see Table 2.

### Integration into the AAVS1 safe harbor locus

For generation of Mic60-TO cells, the donor plasmids pTRE-Tight-Mic60 and AAVS1-SA-2A-NEO-CAG-RTTA3 were co-transfected with the plasmid PX330-AAVS1 into Mic60-KO HeLa cells. Starting 2 days after transfection, cells were selected with 1.25–1.5 µg/ml puromycin (InvivoGen, San Diego, CA, USA) for 3 days. Cells were cultivated without antibiotics for 1 day and afterward selected with DMEM containing 800 µg/ml G418 (Carl Roth, Karlsruhe, Germany) for 8 days. After a recovery period of 10 days, single-cell clones were obtained using a BD Influx cell sorter (BD Biosciences). After clonal expansion, positive clones were detected by PCR and analyzed for Mic60 expression by Western blotting and immunofluorescence staining using a specific antiserum against Mic60 (Proteintech, Rosemont, IL, USA). The plasmid AAVS1-SA-2A-NEO-CAG-RTTA3 was a gift from Paul Gadue (Addgene plasmid #6043; http://n2t.net/addgene:60431; RRID:Addgene_60431). For primers, see Table 3.

To generate Mic10-TO cells, AAVS1-TRE3G-Mic10-FLAG-T2A-EGFP and PX330-AAVS1 were co-transfected into Mic10-KO HeLa cells. Two days after transfection, cells were selected with 1.25–1.5 µg/ml puromycin (Invivogen) for 3 days. After 10 days, cells were induced with 1 µg/ml doxycycline hyclate (Sigma-Aldrich) for 24 h and cells expressing EGFP were sorted using a BD Influx cell sorter (BD Biosciences). After clonal expansion, Mic10-FLAG expression was verified by Western blotting and immunofluorescence staining using specific antisera against the FLAG-tag (Sigma-Aldrich) and Mic10 (Abcam, Cambridge, UK).

Table 1. Oligonucleotides for generation and verification of MICOS-KO cells.

| Oligonucleotides | Sequence (5′–3′) |
| --- | --- |
| gRNA for Mic10-KO FW | CACCGTGTCTGAGTCGGAGCTCGGC |
| gRNA for Mic10-KO REV | AAACGCCGAGCTCCGACTCAGACAC |
| gRNA for Mic13-KO FW | CACCGGCTGGGGGCGCCGTCTACC |
| gRNA for Mic13-KO REV | AAACGGTAGACGGCGCCCCCAGCC |
| gRNA for Mic19-KO FW | CACCGCGAGAATGAGAACATCACCG |
| gRNA for Mic19-KO REV | AAACCGGTGATGTTCTCATTCTCGC |
| gRNA for Mic25-KO FW | CACCGTCTACCTTTGGCCTTCAAGA |
| gRNA for Mic25-KO REV | AAACTCTTGAAGGCCAAAGGTAGAC |
| gRNA for Mic26-KO FW | CACCGTCACTCTACTCAGTTCCTGA |
| gRNA for Mic26-KO REV | AAACTCAGGAACTGAGTAGAGTGAC |
| gRNA for Mic27-KO FW | CACCGACTGCAACTGGTTGTTACAT |
| gRNA for Mic27-KO REV | AAACATGTAACAACCAGTTGCAGTC |
| gRNA for Mic60-KO FW | CACCGCTGCGGGCCTGTCAGTTAT |
| gRNA for Mic60-KO REV | AAACATAACTGACAGGCCCGCAGC |
| Analysis Primer Mic10-KO FW | GGTGAGGAGGAAAGGCCTGGTCACG |
| Analysis Primer Mic10-KO REV | TTCCACTCAAGAGCTCTGCGACTC |
| Analysis Primer Mic13-KO FW | CAGTTCATCAGTTCAAGTGGCGTCCAGCC |
| Analysis Primer Mic13-KO REV | TTACCTGCATTCCAGGAGTCACGGATGG |
| Analysis Primer Mic19-KO FW | GAAAAGAATCCAGGCCCTTCCACGCGC |
| Analysis Primer Mic19-KO REV | CAGTGCCTAGCACTTGGCACAACCAGGAA |
| Analysis Primer Mic25-KO FW | CTCAGCATGGACCTGGTAGGCACTGGGC |
| Analysis Primer Mic25-KO REV | GCCTCAATTCCCACATGGAGAAAGTGGC |
| Analysis Primer Mic26-KO FW | TAAAGTTCAGGTTGCTTGTAACCCTTAGAGTCA |
| Analysis Primer Mic26-KO REV | TATCAAATAGGTTTTATTCATTCTTGCTACTTGC |
| Analysis Primer Mic27-KO FW | CCCCAAAGGATCCATTTTACTGTGGATGGAC |
| Analysis Primer Mic27-KO REV | TCCCAGCTGAACCCAGTCATCCAGCCATCC |
| Analysis Primer Mic60-KO FW | CCTCCGGCAGTGTTCACCTAGTAACCCCTT |
| Analysis Primer Mic60 KO REV | TCGCCCGTCGACCTTCAGCACTGAAAACCTAT |

### Induction of Mic10-FLAG or Mic60 expression in Mic10-TO and Mic60-TO cells

To avoid unintended induction, Mic10-TO and Mic60-TO cells were generally cultivated in DMEM containing tetracycline-free FBS (TAKARA BIO INC., Kusatsu, Japan). To induce expression of Mic10-FLAG or Mic60, the medium was supplemented with doxycycline hyclate (Sigma-Aldrich) at a concentration of 0.025 µg/ml (Mic10-TO cells) or 0.25 µg/ml (Mic60-TO cells) for up to 72 h.

### Transient knockdowns

Knockdowns of Mic10, Mic13, Mic19, Mic25, Mic26, Mic27, Mic60, OPA1, or ATP5ME were achieved by transfection with the respective siRNA pool (siTOOLs Biotech). Cells were cultivated for 2–5 days after transfection.

Knockdown of DRP1 was achieved by transfection with the shRNA expression plasmid pREP4 (Lee et al, 2004). After transfection, cells were selected with DMEM supplemented with 250 µM

**Table 2. Oligonucleotides for generation of plasmids.**

| | |
|---|---|
| pTRE-Tight FW | CCAGAGTGAGATATCTCTAGAGGATCATAATCAGC |
| pTRE-Tight REV | CCGCAGCATGGTGGCGGCGGAATTCTCCAGGCGATCTG |
| Mic60 FW | GCCGCCACCATGCTGCGGGCCTGTCAGTT |
| Mic60 REV | AGAGATATCTCACTCTGGCTGCACCTGAG |
| Mic10-FLAG FW | TCCTACCCTCGTAAAGATATCGCCGCCACCATGTCTGAGTCGGAGCTC |
| Mic10-FLAG REV | CCCTTGCTCACCATGTCGACTGGGCCGGGATTCTCCTCCACGTCACCGCATGTTAGAAGACTTCCTCTGCCCTCACCGGTCTTGTCATCGTCATCCTTG |
| AAVS1-gRNA FW | CACCGTGTCCCTAGTGGCCCCACTG |
| AAVS1-gRNA REV | AAACCAGTGGGGCCACTAGGGACAC |

**Table 3. Oligonucleotides for verification of Mic60-TO cells.**

| | |
|---|---|
| Analyze AAVS1 WT FW | CCCCTATGTCCACTTCAGGA |
| Analyze AAVS1 WT REV | CAGCTCAGGTTCTGGGAGAG |
| Analyze TRE FW | CATTTTTTTCACTGCCTCGACAGTACTAAGC |
| Analyze TRE REV | GAAGGATGCAGGACGAGAAA |
| Analyze CAG FW | TGAATTCACTCCTCAGGTGCAGGCTGCCTAT |
| Analyze CAG REV | GAAGGATGCAGGACGAGAAA |

hygromycin B (Life Technologies, Carlsbad, USA) for 2 days. Afterward, cells were selected for 5 days with 50 μM hygromycin B (Life technologies). All knockdowns were verified by Western blotting and immunofluorescence microscopy.

### Transient expression of Mic10-SNAP

The Mic10-SNAP expression plasmid pH-MINOS1-SNAP was produced by Gateway reaction of pSEMS-GATEWAY-26 m (Covalys Biosciences, Witterswil, Switzerland) and pCR8-MINOS1 (Human ORFeome cDNA clone collection V5.1, Open BioSystems Inc, Huntsville, AL, USA).

### Transient expression of COX8A-SNAP

Transient expression of COX8-SNAP was achieved by transfection with the plasmid AAVS1-Blasticidin-CAG-COX8A-SNAP (Stephan *et al*, 2019).

### Real-time respirometry

Oxygen consumption rate (OCR) experiments were performed in an XF Extracellular Flux Analyzer (Seahorse Bioscience, Billerica, MA, USA) as previously described (Pacheu-Grau *et al*, 2020). Briefly, HeLa cells were seeded at 25,000 cells/well and grown on the Seahorse plate overnight. Baseline respiration was measured in XF DMEM supplemented with 1 mM pyruvate and 10 mM glucose and 2 mM glutamine after incubation at 37°C in an incubator without $CO_2$ for 1 h. Periodic oxygen consumption measurements were performed, and OCR was calculated from the slope of change in oxygen concentration over time. Metabolic states were measured after subsequent addition of 3 μM oligomycin, 1 μM carbonyl cyanide 4 (trifluoromethoxy)phenylhydrazone (FCCP), 1 μM antimycin A, and 2 μM rotenone. For normalization, cell density was calculated using CyQUANT® after OCR measurements, according to manufacturer's instructions by measuring fluorescence intensity (Ex 480 nm, Em 520 nm). OCR values ($N = 6$) were normalized to cell density (ratio of WT) and presented as % of WT.

### Isolation of mitochondria from cultured human cells

Mitochondria were isolated after cell homogenization by differential centrifugation, essentially as previously described (Callegari *et al*, 2016).

### BN–PAGE

Mitochondria were solubilized in 1% digitonin, 20 mM Tris–HCl, pH 7.4, 0.1 mM EDTA, 50 mM NaCl, 10% (w/v) glycerol, 1 mM phenylmethylsulfonyl fluoride for 30 min at 4°C. Unsoluble material was removed by centrifugation at 20,000 $g$ and 4°C for 15 min. After addition of 10× loading dye (5% Coomassie brilliant blue G-250, 500 mM ε-amino n-caproic acid, 100 mM Bis–Tris, pH 7.0), the supernatant was loaded on 4–13% polyacrylamide gradient gels and separated as described before (Wittig *et al*, 2006).

### Affinity purification of protein complexes

For immunoprecipitation of Mic60, the corresponding antibody was coupled to protein A sepharose (GE Healthcare, Chicago, IL, USA) using dimethyl pimelimidate according to the manufactory protocol. WT and MICOS mutant mitochondria were solubilized in a buffer containing 1% digitonin, 20 mM Tris–HCl, pH 7.4, 1 mM EDTA, 100 mM NaCl, 10% (w/v) glycerol, 1 mM phenylmethylsulfonyl fluoride for 1 h at 4°C. Nonsolubilized material was removed by centrifugation at 20,000 $g$ and 4°C for 15 min and the supernatant was mixed with beads. After 1 h binding at 4°C, the beads were washed with 0.3% digitonin buffer containing 20 mM Tris–HCl, pH 7.4, 1 mM EDTA, 100 mM NaCl, 10% (w/v) glycerol, 1 mM phenylmethylsulfonyl fluoride. Bound material was eluted with 100 mM glycine pH 2.8 at room temperature (RT) for 5 min.

For analysis of Mic10-TO cells, whole cells induced with doxycycline hyclate for 8, 16, or 24 h as well as noninduced cells were solubilized in a buffer containing 1% digitonin, 20 mM Tris–HCl, pH 7.4, 1 mM EDTA, 100 mM NaCl, 10% (w/v) glycerol, 1 mM phenylmethylsulfonyl fluoride for 1 h at 4°C. Nonsolubilized material was removed by centrifugation at 20,000 $g$ and 4°C for 15 min. The supernatant was either incubated with FLAG-beads (Sigma-Aldrich) or Mic60-Beads for 1 h at 4°C. The beads were washed with 0.3% digitonin buffer containing 20 mM Tris–HCl, pH 7.4, 1 mM EDTA, 100 mM NaCl, 10% (w/v) glycerol, 1 mM phenylmethylsulfonyl fluoride. Bound material was eluted with 100 mM glycine, pH 2.8 at RT for 5 min.

For an overview on antibodies used, see Table 4.

### Sample preparation for fluorescence microscopy

For immunolabeling, cells were cultured on coverslips for 1–2 days at 37°C with 5% $CO_2$ and fixed with prewarmed (37°C) 4 or 8% formaldehyde in PBS (137 mM NaCl, 2.68 mM KCl, and 10 mM $Na_2HPO_4$, pH 7.4) for 5–10 min at RT. Fixed cells were extracted with 0.5% (v/v) Triton X-100 in PBS, blocked with 5% (w/v) BSA in PBS, and incubated with diluted primary antibodies against Mic60 (Proteintech), Mic19 (Atlas Antibodies), TOM20 (Santa Cruz Biotechnology, Dallas, TX, USA), ATPB (Abcam, Cambridge, UK), FLAG (Sigma-Aldrich), or dsDNA (Abcam) in 5% (w/v) BSA in PBS for 1 h at RT. After washing in PBS, the primary antibodies were detected with secondary goat anti-rabbit or sheep anti-mouse antibodies labeled with Alexa Fluor 594 (Thermo Fisher Scientific) or custom-labeled with Abberior STAR RED (Dye: Abberior, Goettingen, Germany; antibody: Jackson Immuno Research Laboratories, West Grove, PA, USA) in 5% (w/v) BSA in PBS for 1 h at RT. After washing with PBS, the cells were mounted in Mowiol with 0.1% 1,4-Diazabicyclo[2.2.2]octan (DABCO) and 2.5 μg/ml 4′,6-Diamidin-2-phenylindol (DAPI) (Sigma-Aldrich). For MINFLUX nanoscopy, the fixed and extracted cells were incubated with Mic60 antibodies directly labelled with Alexa Fluor 647 (Thermo Fisher Scientific).

For live-cell imaging of COX8A-SNAP or Mic10-SNAP fusion proteins, cells were seeded in glass-bottom dishes (Ibidi GmbH, Martinsried, Germany) before the measurements. Cells were stained with DMEM containing 1 μM SNAP-Cell SiR (New England Biolabs) and 0.1% (v/v) Quant-IT PicoGreen dsDNA reagent (Thermo Fisher Scientific) for 15–20 min. The staining solution was removed, and the cells were washed with DMEM twice. Cells were left in the incubator for about 20 min to remove unbound dye. For imaging, the DMEM was replaced with live-cell imaging solution (Thermo Fisher Scientific). Cells were recorded by stimulated emission depletion (STED) microscopy. For live-cell imaging of mitochondrial membranes, cells were seeded in glass-bottom dishes (Ibidi GmbH, Martinsried, Germany) and stained with DMEM containing 125 nM Mitotracker Green (Thermo Fisher Scientific) for 15–20 min. Cells were washed twice with DMEM and incubated for about 15 min to remove unbound dye. Medium was replaced by live-cell imaging solution (Thermo Fisher Scientific), and cells were recorded with 3D linear structured illumination microscopy (3D SIM). For STED nanoscopy of Mic60 together with COX8A-SNAP, cells expressing COX8A-SNAP were stained with SNAP-cell SiR as described above and subsequently fixed by adding 2xPHEM buffer supplemented with 4.8% formaldehyde and 0.2% glutaraldehyde in equal amounts to the culture medium for 25 min at room temperature. Samples were permeabilized with 0.05% (v/v) Triton X-100 in PHEM buffer for 5 min. To remove free glutaraldehyde, the samples were incubated in 0.1 M ammonium chloride in PHEM for 1 min and afterward were blocked with PHEM containing 1% BSA and 0.2% saponin for 10 min. For immunolabeling, the primary antibody was diluted in blocking solution and incubated overnight at 4°C. Washing was performed five times with blocking solution. Secondary antibodies were diluted in blocking solution and applied for 2 h at room temperature. The sample was washed with blocking solution five times and imaged in PHEM buffer.

### Light microscopy

Confocal microscopy was performed with a TCS SP8 microscope (Leica, Wetzlar, Germany).

STED nanoscopy was performed using dual-color STED 775 QUAD scanning microscopes (Abberior Instruments, Göttingen, Germany) with either a 775 nm Katana-08 HP laser (Onefive GmbH, Regensdorf, Switzerland) or a 775 nm STED-Laser from Abberior Instruments. In brief, for immunolabeled samples the fluorophore Alexa Fluor 594 was exited at 561 or 594 nm and Abberior STAR RED was exited at 640 nm. STED was performed at 775 nm. Images were recorded with a pixel size of 15–20 nm in the 2D STED mode and with a voxel size of 50 nm in the 3D STED mode. For live-cell STED nanoscopy, SNAP-cell SiR was excited at 640 nm and depletion was performed at 775 nm. Images were recorded with a pixel size of 20–25 nm. EGFP or PicoGreen was exited at 488 nm and recorded in the confocal mode. The used objective was an UPlan-SApo 100×/1.40 Oil [infinity]/0.17/FN26.5 objective (Olympus, Tokyo, Japan).

3D structured illumination microscopy of living HeLa cells was performed with a Deltavision OMXv4.0 BLAZE microscope (GE Healthcare, Amersham, UK) using a 60×, 1.42 NA oil immersion PlanApoN objective lens (Olympus) and sCMOS cameras. MitoTracker Green was excited at 488 nm and the emission recorded at 504–552 nm. The intensities and exposure times were set to obtain satisfactory signal strength. A sequence of 15 images for each axial plane, obtained at three different angles with five phases each, was acquired. Multiple axial planes encompassing the entire cell from top to bottom were recorded at a separation of the individual axial planes of 125 nm.

3D MINFLUX nanoscopy was performed with a custom-built MINFLUX nanoscope that was described previously (Gwosch et al,

**Table 4.  Antibodies used for Western blot analysis.**

| Epitope | Source |
|---|---|
| Mic10 | Abcam (Cambridge, UK) and (Callegari et al, 2019) |
| Mic13 | Sigma-Aldrich and (Callegari et al, 2019) |
| Mic19 | Atlas Antibodies (Bromma, Sweden) and (Callegari et al, 2019) |
| Mic25 | Proteintech (Rosemont, IL, USA) and (Callegari et al, 2019) |
| Mic26 | Thermo Fisher Scientific and (Callegari et al, 2019) |
| Mic27 | Atlas Antibodies and (Callegari et al, 2019) |
| Mic60 | Proteintech and (Callegari et al, 2019) |
| ATPB | Molecular Probes (Eugene, OR, USA) |
| ATP5B | Callegari et al (2019) |
| ATP5A | Abcam |
| ATP5ME | Proteintech |
| OPA1 (D7C1A) | Cell Signaling Technology (Danvers, MA, USA) |
| DRP1 | BD Biosciences (San Jose, CA, USA) |
| MFN1 (D6E2S) | Cell Signaling Technology |
| MFN2 (D2D10) | Cell Signaling Technology |
| RIESKE | Callegari et al (2019) |
| COX1 | Callegari et al (2019) |
| LETM1 | Callegari et al (2019) |
| SDHA | Callegari et al (2019) |
| NDUFA10 | Callegari et al (2019) |

2020). For active stabilization of the sample, coverslips were covered with gold nanorods (Nanopartz Inc., Loveland, CO, USA). Nanorods were diluted 1:3 in single molecule clean PBS buffer (Sigma-Aldrich) and sonicated for 5–10 min. The samples were incubated with the nanorod solution for 5–10 min at RT. The sample was washed three times with PBS. For MINFLUX imaging, a standard enzyme-based (d) STORM blinking buffer containing 50 nM Tris/HCl pH 8.0, 10 mM NaCl, 10% (w/v) glucose, supplemented with 0.4 mg/ml glucose oxidase (Sigma-Aldrich) and 90 mM cysteamine hydrochloride (Sigma-Aldrich), was used. Samples were sealed using Picodent Twinsil speed 22 (Picodent Dental-Produktions- und Vertriebs-GmbH, Wipperfürth, Germany). The imaging was performed with parameters and imaging schemes as reported previously (Gwosch *et al*, 2020). Briefly, before MINFLUX measurements, the fluorophore Alexa Fluor 647 was transferred into a long-lived nonfluorescent state by excitation at 642 nm wavelength. Conditional photo-activation of single molecules in the MINFLUX region was performed by illumination at 405 nm wavelength. For localization, molecules were excited with displaced Gaussian or 3D doughnut-shaped excitation beams at 642 nm wavelength. Fluorescence photons were collected in a confocal detection. Scanning of the activation laser and the MINFLUX targeted coordinate pattern was performed in steps of 250 nm in a custom-shaped region selected based on fluorescence widefield images.

**Image processing and analysis**

### Image processing for confocal and STED microscopy

Unless stated in the figure legend, image raw data were not deconvolved, but smoothed with a low-pass filter using the Imspector Software (Abberior Instruments). When deconvolution was applied, we relied on the Richardson-Lucy algorithm and the Imspector software (Abberior Instruments).

In all images, the color tables were adapted for optimal contrast. Background subtraction was usually below 5% of the maximum signal intensity.

### Image processing for 3D SIM

Super-resolved fluorescence images were reconstructed with the corresponding recorded optical transfer function (OTF) in the softWoRx 7.0.0 software (GE Healthcare, Amersham, UK) at a Wiener filter setting of 0.006.

### Image processing for MINFLUX nanoscopy

Data analysis and evaluation were performed as described previously (Gwosch *et al*, 2020). False-positive localizations due to reaction to background or molecular emission events far outside the MINFLUX region were removed based on $p_0 < 0.11$ and $r_{relative} < 32$ nm. To take into account only events from single molecules (i.e., not from groups of simultaneously activated molecules) localizations with high photon count rates (> 100 kHz) were discarded. Localizations with photon numbers above 1,000 and an estimated signal-to-background value larger than 0.6 were selected to guarantee high localization precisions.

### Cluster analysis on STED images of OPA1-depleted cells

Analysis of the clustered fraction of Mic60 was performed by a custom-written MATLAB script. Every image was filtered with a Laplacian of Gaussian (80 nm FWHM) filter and segmented with a threshold of 4% of the brightest value. Segments with an area smaller than 0.0225 $\mu m^2$ were defined as single clusters. The clustered fraction is the ratio of the area of all single cluster segments relative to the total segmented area.

### Sample preparation for electron microscopy

Aclar disks were punched with 18 mm diameter using 0.198 mm thick aclar film (Plano, Wetzlar, Germany) and sterilized with 70% ethanol before usage. Cells were grown on aclar disks to a confluency of ~70% and fixed by immersion using 2% glutaraldehyde in 0.1 M cacodylate buffer at pH 7.4 for 1 h at RT. Fixation was completed overnight at 4°C. After post-fixation in 1% osmium tetroxide and pre-embedding staining with 1% uranyl acetate, samples were dehydrated and embedded in Agar 100 resin (Plano, Wetzlar, Germany). For FIB-SEM, cells were grown on 6 × 0.16 mm sapphire disks (Wohlwend GmbH, Sennwald CH) and vitrified using a Leica EM HPM100 high-pressure freezer (Leica Mikrosysteme Vertrieb GmbH, Wetzlar, Germany). The frozen samples were transferred to an automatic freeze substitution unit Leica EM AFS2 (Leica Mikrosysteme Vertrieb GmbH) and substituted at −90°C for 4 h in a solution containing anhydrous acetone, 2% osmium tetroxide (EMS Electron Microscopical Science, Ft. Washington, USA), 0.1% uranyl acetate in acetone and 5% dest. $H_2O$. After gradually warm up to 0°C, samples were washed with acetone and embedded using Durcupan resin (Science Services GmbH, München, Germany).

### Transmission electron microscopy and electron tomography

Ultrathin sections of ~ 70 nm thickness were recorded on a Philips CM 120 BioTwin transmission electron microscope (Philips Inc., Eindhoven, the Netherlands) without counterstaining. Sections were taken in parallel to the growth surface of the cells. Usually, 2D images of at least 100 mitochondria from at least 10 different cells were randomly taken for each sample, using a TemCam 224A slow scan CCD camera (TVIPS, Gauting, Germany).

For ET, tilt series from 210-nm-thick sections from Agar100 embedded cells were recorded on a Talos L120C transmission microscope (Thermo Fischer Scientific/FEI company, Hilsboro, Oregon, USA) at 17,500× magnification using a Ceta 4k × 4k CMOS camera in unbinning mode. Orthogonal series were recorded from −64.5° to 64.5° using 3° saxton angular increase. The series were calculated using Etomo (David Mastronade, http://bio3d.colorado.edu/). Tomograms were processed using the nonlinear anisotropic diffusion (NAD) filter in IMOD (David Mastronade, http://bio3d.colorado.edu/imod/). Recordings of thin sections were processed in Fiji using the median filter.

### Focused ion beam scanning electron microscopy

Polymerized samples were trimmed with a razor blade, removing empty resin. The sapphire disks were removed, and the tip of the block containing the cells was sawed off with a jigsaw (Villinger *et al*, 2012). The blocks were attached to the SEM stub (Science Services GmbH, Pin 12.7 mm × 3.1 mm) by a silver filled epoxy (Epoxy Conductive Adhesive, EPO-TEK EE 129-4; EMS) and polymerized at 60° overnight. The samples were coated with a 10 nm gold, platinum, or platinum/palladium layer using the sputter coating machine EM ACE600 (Leica Mikrosysteme Vertrieb GmbH) at 35 mA current. The samples were placed into the Crossbeam 540 focused ion beam

scanning electron microscope (Carl Zeiss Microscopy GmbH, Oberkochen, Germany). The SmartSEM software (Carl Zeiss Microscopy GmbH) was used to deposit a 400 nm platinum layer on top of the region of interest using a 3 nA current, to ensure even milling, and to protect the surface. Then, a trench was milled to expose a cross section through the cell of interest using 15 nA current. The cross section was polished using a 7 nA current. The Atlas 3D (Atlas 5.1, Fibics, Canada) software was used to collect the 3D data. The images were acquired at 1.5 kV (analytic mode) with the ESB detector (450 V ESB grid, pixel size $x/y$ 5 nm) in a continuous mill and acquire mode. A 700 pA current was applied to remove 5 nm in between every image. Data post-processing steps were performed in Fiji (Schindelin *et al*, 2012). Image alignments were done using the "Linear Stack Alignment SIFT". The dataset was cropped, inverted, a Gaussian blur (1), and a local contrast enhancement (CLAHE; block-size 127, histogram bins 100, maximum slope 1.5) was applied.

### Segmentation and 3D animation of FIB-SEM and ET data

FIB-SEM and ET data sets were segmented using the software package IMOD. 3D reconstructions of ET data were animated using Amira for Life Sciences (Thermo Fisher Scientific). Reconstructions of FIB-SEM data were animated using Blender (Blender Foundation, Amsterdam, the Netherlands).

### Visualization of MINFLUX data

Data were visualized using Imaris (Bitplane, Belfast, UK).

## Data availability

The authors declare that there are no primary datasets and computer codes associated with this study.

**Expanded View** for this article is available online.

## Acknowledgements

We thank Jan Keller-Findeisen for support with data analysis, Rita Schmitz-Salue for excellent technical assistance, and Jaydev Jethwa for a careful reading of the manuscript. This work was supported by the Deutsche Forschungsgemeinschaft (DFG, German Research Foundation) under Germany's Excellence Strategy—EXC 2067/1-390729940 and by the European Research Council (ERCAdG No. 835102) (to SJ). It was funded by the DFG-funded FOR2848 (project P08 to WM and Z01 to DR) and SFB1190 (project P01 to SJ and P13 to PR).

## Author contributions

SJ and TS conceived the project. TS, CB, MD, PR, DR, and SJ designed research. TS, CB, MD, AMS, MB, TSB, GH, WH, FL, DP-G, and SS performed research. TS, CB, MD, AMS, FB, TSB, PI, JP, TH, SWH, WM, PR, DR, and SJ analyzed data. TS and SJ wrote the paper with comments from all authors.

## Conflict of interest

The authors declare that they have no conflict of interest.

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
