## [Review Process File · The EMBO Journal]

MICOS assembly controls mitochondrial inner membrane remodeling and crista junction redistribution to mediate cristae formation

Stefan Jakobs, Till Stephan, Christian Brüser, Markus Deckers, Anna Steyer, Francisco Balzarotti, Tiana Behr, Gudrun Heim, Wolfgang Hübner, Peter Ilgen, Felix Lange, Jasmin Pape, Stefan Stoldt, Thomas Huser, Stefan Hell, Wiebke Möbius, Peter Rehling, Dietmar Riedel, Mariam Barbot, and David Pacheu-Grau

DOI: [10.15252/emboj.2019104105](https://doi.org/10.15252/emboj.2019104105)

Review Timeline:

Submission Date:	25th Nov 19
Editorial Decision:	7th Jan 20
Revision Received:	17th Mar 20
Editorial Decision:	23rd Apr 20
Revision Received:	8th May 20
Accepted:	13th May 20

Editor: Elisabetta Argenzio

Transaction Report:

Thank you for submitting your manuscript entitled "Mitochondrial inner membrane remodeling and crista junction redistribution drive cristae formation" [EMBOJ-2019-104105] to The EMBO Journal. Please accept my apologies for the delay in communicating our decision due to the recent seasonal holidays. Your study has been sent to three referees for evaluation, whose reviews are enclosed below.

As you can see, the referees find your study interesting and raise a few points that have to be addressed before they can support the publication of your work in The EMBO Journal. In particular, referee #1 requests you to rephrase some statements and to change the title of the manuscript so that it reflects the role of MICOS (sub)complex in cristae structure biogenesis and morphology maintenance. Similarly, reviewer #2 gives you suggestions as to how to improve the manuscript. Finally, referee # 3 asks you to provide co-staining of Mic60 with Cox8a, to perform immunoblot analysis of the holo-MICOS complex and to knock down Mfn1 in order to prove that fission and fusion are not essential for the generation of lamellar cristae.

Given the overall interest of your study, I would like to invite you to revise the manuscript in response to the referee reports. I should note that conclusively addressing these and all the other referees' points is essential for publication in The EMBO Journal.

When preparing your letter of response to the referees' comments, bear in mind that this will form part of the Review Process File and will be available online to the community. For more details on our Transparent Editorial Process, please visit our website: http://emboj.embopress.org/about#Transparent_Process.

We generally grant three months as standard revision time. As a matter of policy, competing manuscripts published during this period will not negatively impact on our assessment of the conceptual advance presented by your study. Nevertheless, please contact me as soon as possible upon publication of any related work.

I thank you again for the opportunity to consider this study for publication and will be happy to answer any questions about the submission of the revised manuscript to The EMBO Journal. I look forward to your revision.

Referee #1:

Stephan et al., have used CRISPR/CAS 9 technology together with a range of imaging modalities to investigate the role of the complete MICOS complex and the Mic-60,19,25,27 sub-complex in cristae structure and biogenesis. The authors were able show quite convincingly that the complete MICOS complex and the sub-complex had different roles in cristae biogenesis and morphology maintenance. On the whole, the manuscript is clear and well presented but I found some of the arguments a little difficult to follow. I think the manuscript would benefit considerably from a clear

statement at the end saying how the complete MICOS and the sub-complex differ in their roles in cristae biogenesis and a model of their working hypothesis shown as a figure in the main article.

Point-by-point

The title needs to be more specific and reflect the manuscript is about the role of the MICOS full and sub-complex in cristae structure.

Summary has a nice statement "We show that the Mic60-subcomplex is sufficient for CJ formations, whereas the Mic10-subcomplex modulates cristae shaping. OPA1 stabilizes tubular CJs and along with the F1Fo ATP synthase, fine-tunes the positioning of MICOS and of the CJs." I like this statement but would appreciate it if it was discussed more in the conclusion section and the authors state more clearly which experiments allowed them to draw these conclusion. E.g. an experimental overview. This would be in contrast to the results section which spells out each experiment in turn.

Figure 1D/S1B. I find these figures difficult to follow. In order to conclude "the existence of Mic60 clusters in the absence of properly developed cristae", wouldn't it be better to look at the cristae shape as shown in 1A using Cox8A with the Mic60 imaging?

Figure 1E. It is not 100% clear what the difference between Total and Eluate is? Also why are there double bands observed for Mic60 in the Total but not eluate? Why is the double band for Mic25 stronger for the Eluate than the Total? Why do mic19-KO mic25-KO only have a single band for Mic25? I notice a double asterisk by the lower band in Mic25? is this because Mic25 also binds Mic19 which is why you have the double band? The meaning of the double asterisk needs to be stated on the figure or in the legend.

Figure 1F. A comment on the respiratory chain supercomplex assembly would be appreciated.

Figure 1G/H. What is the criteria used to determine whether something is tubular, intermediate or fragmented. For 1G, a zoom in on one area of the mitochondria to see the different morphologies would be appreciated.

Figure 2F, is it possible to provide zoomed in pictures of the cristae junctions, specifically for Mic10-KO.

Page 8, line 3 of new section. "opposing" what do the authors mean when they say opposing? This phrase is also used on page 10, 4th line from page end.

Page 9 and figure 4C-D. The authors state that they want to answer the question as to whether cristae in Mic60-KO cells are converted to the WT morphology or whether they are replaced by newly formed cristae. To answer this question they used electron tomography. Electron tomography just provides a snap shot of the state of the cristae at a specific time point. It can only give one time point per mitochondria as the sample has to be dead when imaged. Thus, you can't assess whether new cristae are formed or existing cristae are remodeled using this technique. You could do it with live cell imaging e.g. PMID 31337683. Please can the authors rephrase the paragraph to reflect that electron tomography gives insight into a single time point and each image is from a different cell and the changes which happen to a single cristae can not be followed by this technique. Alternatively, please add live cell imaging data of the quality presented in PMID 31337683.

Figure 4F, page 10. How do the authors determine that "a substantial part of the CJs induced by Mic60 expression was found on aberrant or intermediately shaped cristae" especially as tomography gives only a single insight into a cell?

Please add a schematic figure to the main article describing how the MICOS sub-complex and halo-complex function in cristae biogenesis. Specifically, the location of the sub complex and whole complex should be added to the images on the right hand panel of figure S8. For the left side of figure S8, how are we looking at the cristae? is it a longitudinal section through the cristae?

Spelling errors/suggestions.

Page 4: 1st paragraph of results section, line 9. I would change 'length' to 'long'.

Page 7: Second paragraph, line 4. Please change 'und' to 'and'.

Page 8: line 5. Please add two commas: "In such mitochondria, we recorded, only occasionally, very few CJs" Alternatively rephrase to make sentence clearer.

Page 11: 3rd paragraph, line 2: please change "FIB-SEB" to "FIB-SEM"

Figure 5C. middle bottom panel. Please remove "s" from "(high exspression)".

Figure 7D. please change orange label from "versicular" to "vesicular"

Referee #2:

In this manuscript, Stephan and colleagues elegantly dissected the requirements for MICOS complexes using KO lines coupled with exquisite image analysis. This included super-resolution approaches, FIB-SEM and reconstruction of cristae junctions and cristae lamellae within mitochondrial sections.

The authors establish that the Mic60-subcomplex, which is stable in the absence of the Mic10-subcomplex, is essential for the maintenance of CJs and the stability of the holo-MICOS complex. The Mic10-subcomplex is essential for lamellar cristae formation. Through knockdown studies, the authors established an important role of OPA1 for stabilizing the presence of tubular cristae junctions in mitochondrial lacking Mic10. This suggests that both Mic10 subcomplexes and Opa1 play complementary roles in cristae junction stability.

Their findings lead to a new model for the formation of cristae in higher eukaryotes that is nicely depicted in Fig. S8.

Overall this is an excellent study and am happy to see it published soon and in its current form. Suggestions below are worth consideration by the authors but in my opinion, they are not essential. This was a pleasure to read and will be highly valued by the community.

Comments

1. Given the lack of a complete Mic60-KO, the authors could provide a table noting the genotype results of indels in the alleles targeted in each knockout clone and the region targeted.

2. The authors conclude that OXPHOS assembly is largely independent from MICOS formation (Fig 1F) yet the supercomplex is impaired in the Mic60-KO line. The authors have not addressed

complexes I and II and this could be performed since it is not entirely convincing.

3. Zoomed images of mitochondrial morphology changes in Fig 1G would be useful.

4. Drp1 essentiality has been shown by groups while Fonseca et al. refers to the fact that dynamin-2 is not required so this cite is inappropriate.

5. Given that changes in cristae/Mic60 location following loss of ATP5ME mirrors that of Mic10 KO cells the authors could confirm that the Mic10 subcomplex components are still stably expressed in these cells.

6. Fig S8 is a nice depiction of the cristae assembly model. This would be nicely placed in the main body of the article (or alternately as part of a N&V piece).

Referee #3:

This paper investigates the relationship between MICOS subcomplexes, Opa1, and ATP synthase dimers in cristae formation and stability. The beautiful superresolution images and electron tomographic reconstructions are in line with several other previously published reports, acknowledged by the authors, dissecting the effect of the regulatory cristae biogenesis proteins in mammalian cells. As a matter of fact, the manuscript does not contain a formal epistatic analysis and hence it is falling short of reaching the level of definition of the previous papers cited by the authors. This shortcoming is however greatly compensated by the technical tour de force and by the detailed analysis of the role of each MICOS subunit, as well as by the novelty of the message on the role of Mic10.

Authors show that upon Mic10 ablation the Mic10 but not the Mic60 MICOS subcomplex is lost, whereas Mic60 ablation leads to the loss of all MICOS subunits. They also show that the Mic10 subcomplex is recruited to Mic60 subcomplex, and that the Mic60 subunits bind to Mic60 independently of Mic10 subcomplex. This system allows them to separate the role of Mic10 subcomplex from holo-MOCOS complex. They characterise the phenotype of each KO and show that Mic60 and Mic10 display the strongest phenotypes compared to Mic13 and Mic19. They nicely show that rescue of Mic10 leads to an intermediate phenotype, suggesting remodelling of aberrant cristae rather than biogenesis of new cristae. All in all, their results demonstrate that MICOS seems to not be required for cristae formation, but that Mic60 is critical for CJ formation and/or maintenance, and Mic10 is required for cristae organisation.

In the last part of the manuscript, authors investigate the relationship between MICOS, Opa1 and ATP synthase dimers in the organization of the cristae. They conclude that Opa1 regulates Mic60 oligomers assembly/stability and hence stability of the CJ, similar to the effect recorded upon disruption of dimers of the ATP synthase. The following points must be addressed.

Major concerns

1. Fig .1C, D; The authors conclude that Mic60 clusters can be found in cristae-free voids. The images show costaining of Cox8A with DNA, and of Mic60 with DNA. To conclude that Mic60 clusters can be found in cristae-free voids, a costaining of Mic60 with Cox8A is required.

2. Fig. 2F. Here authors show tomograms of WT, Mic10-KO and Mic60-KO mitochondria. The chosen mitochondria from Mic10-KO seems a "2 membranes" or an "arc-shaped" mitochondria whereas the one of Mic60-KO seems to be an "onion-shaped" mitochondria, according to fig.2C. It would be interesting to add similar reconstruction of the same type of architecture for both genetic

background: e.g. one "arc-shaped" Mic60-KO and one "onion-shaped" Mic10-KO. This would allow to compare the difference of Mic10 vs Mic60 knockouts regardless the architecture of the mitochondria, and demonstrate that the differences in CJs observed here is not specific to a mitochondrial architecture.

3. The authors conclude that Mic60 is differently distributed in Mic10-KO cells compared to WT. In addition, a different distribution of Mic60 in different conditions is shown multiple times. However, these results are not very informative. Indeed, it is difficult to understand if this distribution is a consequence of the aberrant mitochondrial architecture of Mic10-KO cells, or it is due to anormal location of Mic60. Mic10-KO cristae are indeed close to and parallel to the OM, with tubular CJs connecting them to the IBM. Thus, it is not clear if Mic60 is mislocated within mitochondria (e.g. on the IBM) or if this staining pattern is due to the aberrant architecture, with no difference of localisation compared to the WT.

4. It is not clear if Mic10 can influence MICOS localisation. Perhaps Mic60 staining in e.g. Mic19-KO that still express Mic10 could answer if the different distribution is a consequence of the aberrant architecture of Mic10-KO cells, or if Mic10 directly regulates MICOS localisation. Similarly, distribution of Mic60 is affected by OPA1-KD, Mic60 distribution is differently regulated upon OPA1-KD in WT and Mic10-KO cells.

Fig.S5. Authors conclude that upon re-expression of Mic10 the Mic60-IP show that "the Mic10-subcomplex proteins bind to the pre-existing Mic60-subcomplex to form the holo-MICOS complex". However, immunoblotting of Mic25 and Mic27 are missing to show the holo-MICOS complex. In addition, no blue native gel proving holo-MICOS complex assembly is shown.

FigS7. Upon DRP1 KD authors conclude that both fission and fusion are not essential for lamellar cristae generation. KD of e.g. Mfn1 is required to support this conclusion.

Minor concerns

There is a mistake in Fig.S1C: the 2 first lanes are "Mic10-KO" and no WT is indicated as in the legend. In addition, levels of Mic10 in the first lane is OK, so I guess this is the WT lane.

The authors show that OPA1-KD in Mic10-KO cells reduce the number of CJ. It would be interesting to check the size of the remaining CJ. It would be also interesting to check if OPA1 distribution is changing in Mic10-KO and Mic60-KO cells.

It would be interesting to check mitochondrial function in the Mic-KO cells. Blue native for complexes III, IV and V has been performed, but other experiments e.g. respiration would be interesting to correlate aberrant mitochondria ultrastructure caused by the different MICOS subunit ablation with mitochondria dysfunction.

Videos captions and numbers are missing.

Point-by-point response

Reviewers' Comments:

Referee #1:

Stephan et al., have used CRISPR/CAS 9 technology together with a range of imaging modalities to investigate the role of the complete MICOS complex and the Mic-60,19,25,27 sub-complex in cristae structure and biogenesis. The authors were able show quite convincingly that the complete MICOS complex and the sub-complex had different roles in cristae biogenesis and morphology maintenance. On the whole, the manuscript is clear and well presented but I found some of the arguments a little difficult to follow. I think the manuscript would benefit considerably from a clear statement at the end saying how the complete MICOS and the sub-complex differ in their roles in cristae biogenesis and a model of their working hypothesis shown as a figure in the main article.

We thank the referee for her/his positive view on our manuscript.

We followed the suggestion of providing a clear statement on the role of MICOS and the two sub-complexes by rewriting the entire discussion section.

Importantly, the new main table (Fig 10D) provides a concise overview on the key observations and the new main Fig 10A provides a model of the working hypothesis.

The title needs to be more specific and reflect the manuscript is about the role of the MICOS full and sub-complex in cristae structure.

We carefully discussed this concern and changed the title of our manuscript accordingly. It now reads:

“MICOS assembly controls mitochondrial inner membrane remodeling and crista junction redistribution to mediate cristae formation“. (It was: *“Mitochondrial inner membrane remodeling and crista junction redistribution drive proper cristae formation”*)

We believe that the new title reflects the content of the manuscript more appropriately.

Summary has a nice statement "We show that the Mic60-subcomplex is sufficient for CJ formations, whereas the Mic10-subcomplex modulates cristae shaping. OPA1 stabilizes tubular CJs and along with the F1Fo ATP synthase, fine-tunes the positioning of MICOS and of the CJs." I like this statement but would appreciate it if it was discussed more in the conclusion section and the authors state more clearly which experiments allowed them to draw these conclusion. E.g. an experimental overview. This would be in contrast to the results section which spells out each experiment in turn.

We thank the referee for this suggestion. To accommodate it, we re-wrote the entire discussion section and added the new Fig 10, which shows a summary of the experimental findings and a cartoon depicting our model of crista biogenesis, which is based on these findings.

We believe that this new figure clarifies our findings and benefits the readability of the manuscript.

Figure 1D/S1B. I find these figures difficult to follow. In order to conclude "the existence of Mic60 clusters in the absence of properly developed cristae", wouldn't it be better to look at the cristae shape as shown in 1A using Cox8A with the Mic60 imaging?

We followed this suggestion and performed the experiment. The new panel (Fig. 1E) shows a dual-color staining of COX8A-SNAP and Mic60. It fully supports our initial conclusion. We agree that this a direct way to prove "the existence of Mic60 clusters in the absence of properly developed cristae".

Figure 1E. It is not 100% clear what the difference between Total and Eluate is? Also why are there double bands observed for Mic60 in the Total but not eluate? Why is the double band for Mic25 stronger for the Eluate than the Total? Why do mic19-KO mic25-KO only have a single band for Mic25? I notice a double asterisk by the lower band in Mic25? is this because Mic25 also binds Mic19 which is why you have the double band? The meaning of the double asterisk needs to be stated on the figure or in the legend.

We tried to clarify all these issues in the revised manuscript. We explain that the total is the fraction of isolated and digitonin-permeabilized mitochondria that were added to the beads, whereas the eluate is the fraction that was obtained from the beads after several washing steps.

The double-band of Mic60 in the total fraction is due to unspecific binding of the antibody, as it is not present in the eluate. All this should be clearer in the carefully revised new version of the manuscript

We now explain explicitly that the double-band of Mic25 is due to the fact that the antibody detects also Mic19:

Page 39: **** Unspecific band, due to the cross reaction of the anti-Mic25 antibody with Mic19.*

We believe that this explanation answers all subsequent questions raised by the referee on the two bands detected by the Mic25 antibody.

Figure 1F. A comment on the respiratory chain supercomplex assembly would be appreciated.

We added substantial new data on respiratory chain supercomplex assembly to the revised version of the manuscript. The new Fig EV1 shows BN-PAGEs of all five supercomplexes.

It now reads (Page 7): *“Even in the absence of Mic60, virtually resulting in the absence of MICOS, the assembly of complexes I, II, and V was nearly unaffected and the assembly of complexes III and IV was only slightly decreased (Fig EV1D).”*

Figure 1G/H. What is the criteria used to determine whether something is tubular, intermediate or fragmented. For 1G, a zoom in on one area of the mitochondria to see the different morphologies would be appreciated.

We added a magnification to the former Fig. 1G (Fig. 2C in the revised manuscript). Evaluation was performed manually in a double-blind approach based on pre-defined morphology criteria.

It now reads (Page 39): “(D) Quantification of the mitochondrial networks as shown in (C). The evaluation was performed manually in a blinded approach based on pre-defined morphology criteria. Average and SD of 3 independent biological replicates are shown (>170 cells per sample and replicate). Scale bar: 10 μ m.”

Figure 2F, is it possible to provide zoomed in pictures of the cristae junctions, specifically for Mic10-KO.

As we already have 10 main figures with many panels, we are hesitant to add another panel. Instead, we provide in the revised version of the manuscript Movie EV6, which provides a detailed view on the crista junctions.

Page 8, line 3 of new section. "opposing" what do the authors mean when they say opposing? This phrase is also used on page 10, 4th line from page end.

We thank the reviewer for pointing us to this. Meant was “opposite”. This is changed in the revised manuscript. Moreover, we define in the revised manuscript our definition of “opposite distribution bands”:

It now reads (page 9):

“... localized in clearly discernibly opposite distribution bands, i.e. they exhibited a two-sided distribution on the mitochondrial tubules (Fig 4A, Fig EV2A).”

Page 9 and figure 4C-D. The authors state that they want to answer the question as to whether cristae in Mic60-KO cells are converted to the WT morphology or whether they are replaced by newly formed cristae. To answer this question they used electron tomography. Electron tomography just provides a snap shot of the state of the cristae at a specific time point. It can only give one time point per mitochondria as the sample has to be dead when imaged. Thus, you can't assess whether new cristae are formed or existing cristae are remodeled using this technique. You could do it with live cell imaging e.g. PMID 31337683. Please can the authors rephrase the paragraph to reflect that electron tomography gives insight into a single time point and each image is from a different cell and the changes which happen to a single cristae can not be followed by this technique. Alternatively, please add live cell imaging data of the quality presented in PMID 31337683.

Figure 4F, page 10. How do the authors determine that "a substantial part of the CJs induced by Mic60 expression was found on aberrant or intermediately shaped cristae" especially as tomography gives only a single insight into a cell?

We are fully aware that all electron microscopy is performed on fixed cells and thus provides inescapably a snapshot on the cell under investigation.

Still, our approach allows us to get insights into the sequence of events occurring during the recovery of Mic60-TO cells upon Mic60 expression: We demonstrate in this manuscript that upon induction of Mic60 (or Mic10) by doxycycline, the recovery of the crista structures across the cells in a dish is largely simultaneous. Hence, by

recording EM images of many cells at different time points we can indeed follow the recovery of the structures.

To demonstrate that the expression levels and the recovery upon induction by doxycycline are similar from cell to cell at a given time point, we added two new figures to the manuscript (Appendix Fig 3 and Appendix Fig 4).

Appendix Fig 3 shows the simultaneous raise of the expression levels in a cell culture.

Appendix Fig 4 shows a selection of mitochondria from cells induced for 0 h and 16 h.

Without induction, only very few CJs are present and nearly no WT-like cristae. After 16 h of induction, the cristae are mostly of an aberrant morphology. However, the number of CJs is increased and the additional CJs are mostly found on cristae with the KO-phenotype. This clearly suggests that the new CJ are secondary CJs that are formed on already existing crista.

We took great care to explain this approach in the revised manuscript.

Please add a schematic figure to the main article describing how the MICOS sub-complex and halo-complex function in cristae biogenesis. Specifically, the location of the sub complex and whole complex should be added to the images on the right hand panel of figure S8. For the left side of figure S8, how are we looking at the cristae? is it a longitudinal section through the cristae?

We thank the reviewer for these suggestions. We added the new Fig 10 to the manuscript that summarizes our findings and our conclusions. It also contains a model for cristae formation and for the roles of both MICOS-subcomplexes. We took great care to carefully explain the items shown in the figure and hope that we could address all concerns by the reviewer on the previous Fig S8.

Spelling errors/suggestions.

Page 4: 1st paragraph of results section, line 9. I would change 'length' to 'long'.

Page 7: Second paragraph, line 4. Please change 'und' to 'and'.

Page 8: line 5. Please add two commas: "In such mitochondria, we recorded, only occasionally, very few CJs" Alternatively rephrase to make sentence clearer.

Page 11: 3rd paragraph, line 2: please change "FIB-SEB" to "FIB-SEM"

Figure 5C. middle bottom panel. Please remove "s" from "(high exspression)".

Figure 7D. please change orange label from "versicular" to "vesicular"

Done. We corrected all typos.

Referee #2:

In this manuscript, Stephan and colleagues elegantly dissected the requirements for MICOS complexes using KO lines coupled with exquisite image analysis. This included super-resolution approaches, FIB-SEM and reconstruction of cristae junctions and cristae lamellae within mitochondrial sections.

The authors establish that the Mic60-subcomplex, which is stable in the absence of the Mic10-subcomplex, is essential for the maintenance of CJs and the stability of the holo-MICOS complex. The Mic10-subcomplex is essential for lamellar cristae formation. Through knockdown studies, the authors established an important role of OPA1 for stabilizing the presence of tubular cristae junctions in mitochondrial lacking Mic10. This suggests that both Mic10 subcomplexes and Opa1 play complementary roles in cristae junction stability. Their findings lead to a new model for the formation of cristae in higher eukaryotes that is nicely depicted in Fig. S8.

Overall this is an excellent study and am happy to see it published soon and in its current form. Suggestions below are worth consideration by the authors but in my opinion, they are not essential. This was a pleasure to read and will be highly values by the community.

We thank the reviewer for his/her encouraging view on our manuscript.

Comments

1. Given the lack of a complete Mic60-KO, the authors could provide a table noting the genotype results of indels in the alleles targeted in each knockout clone and the region targeted.

Done. We added to the revised manuscript Appendix Table S1 that summarizes the targeted exons and shows sequencing results of about 20 subclones for each KO cell line.

2. The authors conclude that OXPHOS assembly is largely independent from MICOS formation (Fig 1F) yet the supercomplex is impaired in the Mic60-KO line. The authors have not addressed complexes I and II and this could be performed since it is not entirely convincing.

Done. The new Fig EV1D shows BN-PAGEs of all five supercomplexes. In the revised manuscript, we also provide Seahorse data on all KO-cell lines in the new Fig 2B.

Based on these data we conclude:

Page 7: *“Even in the absence of Mic60, virtually resulting in the absence of MICOS, the assembly of complexes I, II, and V was nearly unaffected and the assembly of complexes III and IV was only slightly decreased (Fig EV1D). In Mic60-KO cells, the oxygen consumption rate was reduced, but not abolished; all other MICOS-KO cell lines exhibited oxygen consumption rates close to the WT (Fig 2B). We conclude that the deletion of MICOS subunits has only modest influence on OXPHOS assembly.”*

3. Zoomed images of mitochondrial morphology changes in Fig 1G would be useful.

Done. We added a magnification to the former Fig. 1G (Fig. 2C in the revised manuscript).

4. Drp1 essentiality has been shown by groups while Fonseca et al. refers to the fact that dynamin-2 is not required so this cite is inappropriate.

The reviewer is correct. We corrected our mistake and no longer cite Fonseca et al.

5. Given that changes in cristae/Mic60 location following loss of ATP5ME mirrors that of Mic10 KO cells the authors could confirm that the Mic10 subcomplex components are still stably expressed in these cells.

We thank the reviewer for suggesting this experiment. The data is shown in the new Fig EV5B. The western blot shows that the levels of Mic10, Mic13, Mic26, Mic27 and Mic60 are not affected upon depletion of ATP5ME.

6. Fig S8 is a nice depiction of the cristae assembly model. This would be nicely placed in the main body of the article (or alternately as part of a N&V piece).

We thank the reviewer for suggesting this. We included the model into the entirely new main Fig 10 that also summarizes our results and our conclusions. We would be happy to see our manuscript discussed in a N&V piece.

Referee #3:

This paper investigates the relationship between MICOS subcomplexes, Opa1, and ATP synthase dimers in cristae formation and stability. The beautiful superresolution images and electron tomographic reconstructions are in line with several other previously published reports, acknowledged by the authors, dissecting the effect of the regulatory cristae biogenesis proteins in mammalian cells.

As a matter of fact, the manuscript does not contain a formal epistatic analysis and hence it is falling short of reaching the level of definition of the previous papers cited by the authors. This shortcoming is however greatly compensated by the technical tour de force and by the detailed analysis of the role of each MICOS subunit, as well as by the novelty of the message on the role of Mic10.

Authors show that upon Mic10 ablation the Mic10 but not the Mic60 MICOS subcomplex is lost, whereas Mic60 ablation leads to the loss of all MICOS subunits. They also show that the Mic10 subcomplex is recruited to Mic60 subcomplex, and that the Mic60 subunits bind to Mic60 independently of Mic10 subcomplex. This system allows them to separate the role of Mic10 subcomplex from holo-MOCOS complex. They characterise the phenotype of each KO and show that Mic60 and Mic10 display the strongest phenotypes compared to Mic13 and Mic19. They nicely show that rescue of Mic10 leads to an intermediate phenotype, suggesting remodelling of aberrant cristae rather than biogenesis of new cristae. All in all, their results demonstrate that MICOS seems to not be required for cristae formation, but that Mic60 is critical for CJ formation and/or maintenance, and Mic10 is required for cristae organisation.

In the last part of the manuscript, authors investigate the relationship between MICOS, Opa1 and ATP synthase dimers in the organization of the cristae. They conclude that Opa1 regulates Mic60 oligomers assembly/stability and hence stability of the CJ, similar to the effect recorded upon disruption of dimers of the ATP synthase. The following points must be addressed.

We thank this referee for his/her expert review and the positive assessment of our manuscript.

Major concerns

1. Fig. 1C, D; The authors conclude that Mic60 clusters can be found in cristae-free voids. The images show costaining of Cox8A with DNA, and of Mic60 with DNA. To conclude that Mic60 clusters can be found in cristae-free voids, a costaining of Mic60 with Cox8A is required.

We thank the reviewer for suggesting this. We followed this suggestion and performed the experiment. The new panel (Fig. 1E) shows a dual-color staining of COX8A-SNAP and Mic60. It fully supports our initial conclusion. We agree that this a direct way to prove the existence of Mic60 clusters in the absence of properly developed cristae.

2. Fig. 2F. Here authors show tomograms of WT, Mic10-KO and Mic60-KO mitochondria. The chosen mitochondria from Mic10-KO seems a "2 membranes" or an "arc-shaped" mitochondria whereas the one of Mic60-KO seems to be an "onion-shaped" mitochondria,

according to fig.2C. It would be interesting to add similar reconstruction of the same type of architecture for both genetic background: e.g. one "arc-shaped" Mic60-KO and one "onion-shaped" Mic10-KO. This would allow to compare the difference of Mic10 vs Mic60 knockouts regardless the architecture of the mitochondria, and demonstrate that the differences in CJs observed here is not specific to a mitochondrial architecture.

To address this concern, we show in the revised version of the manuscript the requested tomograms: Movie EV7 shows an "onion-shaped" Mic10-KO mitochondrion and Movie EV4 an "arc-shaped" Mic60-KO mitochondrion.

The tomograms evidence that irrespective of the cristae architecture, CJs are missing in Mic60-KO mitochondria, whereas they can be observed in Mic10-KO cells. Hence, these new data sets demonstrate that differences in CJs are indeed due to the KO and not to a specific mitochondrial architecture.

3. The authors conclude that Mic60 is differently distributed in Mic10-KO cells compared to WT. In addition, a different distribution of Mic60 in different conditions is shown multiple times. However, these results are not very informative. Indeed, it is difficult to understand if this distribution is a consequence of the aberrant mitochondrial architecture of Mic10-KO cells, or it is due to anormal location of Mic60. Mic10-KO cristae are indeed close to and parallel to the OM, with tubular CJs connecting them to the IBM. Thus, it is not clear if Mic60 is mislocated within mitochondria (e.g. on the IBM) or if this staining pattern is due to the aberrant architecture, with no difference of localisation compared to the WT.

In the revised manuscript, we took great care to clarify the line of argumentation.

It now reads:

(Page 10): *“The ET, FIB-SEM and super-resolution data conclusively demonstrate that in Mic10-KO cells the CMs form generally single-layered hollow tubes. It is difficult to reconcile the formation of the opposite distribution bands with a large tube-like CM that evenly lines the mitochondrial tubule. We thus conclude that the distribution of the F1Fo-ATP synthase and of the Mic60 clusters in opposite bands is not primarily determined by the shape of the tube-like CM, but predict that the abundance of Mic10 controls the distribution of Mic60. To test this prediction further,”*

4. It is not clear if Mic10 can influence MICOS localisation. Perhaps Mic60 staining in e.g. Mic19-KO that still express Mic10 could answer if the different distribution is a consequence of the aberrant architecture of Mic10-KO cells, or if Mic10 directly regulates MICOS localisation. Similarly, distribution of Mic60 is affected by OPA1-KD, Mic60 distribution is differently regulated upon OPA1-KD in WT and Mic10-KO cells.

We thank the referee for this insightful suggestion, which helped to significantly improve the manuscript.

As suggested, we determined the sub-mitochondrial distribution of Mic60 in Mic19-KO cells with different expression levels of Mic10 (new Fig 4F and new Fig EV2F). We found that depending on the Mic10 levels, Mic60 exhibits different sub-mitochondrial distributions in Mic19-KO cells. At low Mic10 levels, Mic60 is scattered in clusters over the inner boundary membrane. Upon Mic10 overexpression in Mic19-KO cells, Mic60 forms assemblies (new Fig 4F and new Fig 2F). Remarkably, Mic10 overexpression also stabilized the Mic60 levels in Mic19-KO cells. Interestingly, also

in OPA1-depleted cells, the formation of the Mic60 assemblies is Mic10-dependent (Fig EV4F). The data thus conclusively show that Mic10 regulates the size of Mic60 assemblies.

We write in the revised manuscript:

(Page 10): *“The formation of such continuous Mic60 assemblies was strongly increased when we overexpressed Mic10-FLAG in Mic19-KO cells (Fig 4F, Fig EV2F). In addition, Mic10-FLAG overexpression also raised the expression level of Mic60 in Mic19-KOs (Fig EV2F).*

Altogether, these data demonstrate that the expression level of Mic10 influences the distribution of Mic60 and also of the F1Fo-ATP synthase. In the absence of Mic10, Mic60 is found in clusters localized in opposite distribution bands, whereas at elevated Mic10 levels, Mic60 forms extended assemblies.”

Fig.S5. Authors conclude that upon re-expression of Mic10 the Mic60-IP show that "the Mic10-subcomplex proteins bind to the pre-existing Mic60-subcomplex to form the holo-MICOS complex". However, immunoblotting of Mic25 and Mic27 are missing to show the holo-MICOS complex. In addition, no blue native gel proving holo-MICOS complex assembly is shown.

We performed the requested experiment: The new Appendix Figure S5A shows Co-IPs (using Mic10-FLAG and Mic60 as a bait) decorated for all MICOS subunits including Mic25 and Mic27. The blot demonstrates that the holo-MICOS complex is formed.

FigS7. Upon DRP1 KD authors conclude that both fission and fusion are not essential for lamellar cristae generation. KD of e.g. Mfn1 is required to support this conclusion.

We thank the referee for this valuable suggestion. We depleted Mfn1 (and in addition also Mfn2). TEM recordings demonstrated that in the absences of Mfn1, Mfn2, or both, the mitochondria still exhibited lamellar cristae (new Fig EV4C-D). We therefore conclude that fusion and fission of mitochondrial tubules is not necessary for lamellar cristae formation.

In the revised manuscript it reads:

(Page 14): *”In mammalian cells, the fusion of the mitochondrial OM is regulated by the two mitofusins MFN1 and MFN2, two highly conserved dynamin-related GTPases, which exhibit distinguishable functions (Giacomello, Pyakurel et al., 2020, Ishihara, Eura et al., 2004). To investigate if OM fusion is essential for lamellar crista formation, we depleted HeLa cells for MFN1 or MFN 2 or MFN1 together with MFN2. Depletion of these proteins resulted in a mild cristae phenotype, but lamellar cristae were still observed (Fig EV4C-D). We conclude that in mammalian cells OM fission or fusion are not essential for the development of lamellar cristae.”*

Minor concerns

There is a mistake in Fig.S1C: the 2 first lanes are "Mic10-KO" and no WT is indicated as in the legend. In addition, levels of Mic10 in the first lane is OK, so I guess this is the WT lane.

Thanks for pointing to this error in the labeling of the figure. We corrected this mistake.

The authors show that OPA1-KD in Mic10-KO cells reduce the number of CJ. It would be interesting to check the size of the remaining CJ. It would be also interesting to check if OPA1 distribution is changing in Mic10-KO and Mic60-KO cells.

Absolutely, this is interesting, although it is beyond the scope of this manuscript. We will address these questions in future studies.

It would be interesting to check mitochondrial function in the Mic-KO cells. Blue native for complexes III, IV and V has been performed, but other experiments e.g. respiration would be interesting to correlate aberrant mitochondria ultrastructure caused by the different MICOS subunit ablation with mitochondria dysfunction.

We performed the requested experiments.

The new Fig EV1D shows BN-PAGEs of all five supercomplexes. In the revised manuscript, we also provide respiration (Seahorse) data on all KO-cell lines in the new Fig 2B.

Based on these data we conclude:

(Page 7:) *“Even in the absence of Mic60, virtually resulting in the absence of MICOS, the assembly of complexes I, II, and V was nearly unaffected and the assembly of complexes III and IV was only slightly decreased (Fig EV1D). In Mic60-KO cells, the oxygen consumption rate was reduced, but not abolished; all other MICOS-KO cell lines exhibited oxygen consumption rates close to the WT (Fig 2B). We conclude that the deletion of MICOS subunits has only modest influence on OXPHOS assembly.”*

Videos captions and numbers are missing.

We moved the video caption from the supplement file to the manuscript file.

Thank you for submitting a revised version of your manuscript . Please accept my apologies for the delay in getting back to you with our decision due to a belated report . Your study has now been seen by the original referees whose comments are shown below.

As you will see, they find that all criticisms have been sufficiently addressed and recommend the manuscript for publication pending text modifications. In addition, before we can officially accept the manuscript , there are a few editorial issues concerning text and figures that I need you to address.

REFEREE REPORTS

Referee #1:

Till et al., have made thoughtful revisions to their manuscripts and address the vast majority of my concerns. I was disappointed not to see a sentence stating how the respiratory chain supercomplexes I,III₂,IV_x and III₂/IV₂ are affected by the Micos mutants. The authors did state that complex I, II and V were unaffected and complex III and IV slightly affected. They made no comment on whether the supercomplex I,III₂,IV_x and III₂/IV₂ still exist in the crista membranes and whether the level was similar to WT . I think this would be a very nice addition to the manuscript and make the story complex.

There were a few minor mistakes:

Results 4th line, should the authors also refer to fig 1C?

Page 9 first line. C is missing from Mic60

Page 11 last line, "they" should be clearly stated as to what the authors mean by they.

Page 16 Discussion, 5 lines from bottom of page. The sentence should read "allow us to draw conclusions"

Overall the manuscript is very nice and the work of exceptional high quality which should be published in EMBO.

Referee #2:

As previously noted, I am very excited by the quality and conclusions of this manuscript. I am happy for it accepted although I do request a slight change:

I am not keen on the revised sentence on page 7 :

"Even in the absence of Mic60, virtually resulting in the absence of MICOS, the assembly of complexes I, II, and V was nearly unaffected and the assembly of complexes III and IV was only slightly decreased (Fig EV1D)."

Rather than "nearly unaffected", "somewhat impaired" or similar would be more appropriate. In particular there does seem to be clear changes in the complex III-complex IV super complex profile in the Mic60 "knockout". This should not be dismissed in toto.

Referee #3:

Authors performed an extensive revision of their paper and addressed experimentally and satisfactorily all the points raised. The only missing experiment is the BN-PAGE for the MICOS holocomplex. From the co-IP it is impossible to conclude that the holocomplex is assembled (co-IPs are by definition performed on solubilized material), hence I suggest that before the paper is printed, this conclusion is modified.

Point-by-point response

Reviewers' Comments:

Referee #1:

Till et al., have made thoughtful revisions to their manuscripts and address the vast majority of my concerns. I was disappointed not to see a sentence stating how the respiratory chain supercomplexes I,III2,IVx and III2/IV2 are affected by the Micos mutants. The authors did state that complex I, II and V were unaffected and complex III and IV slightly affected. They made no comment on whether the supercomplex I,III2,IVx and III2/IV2 still exist in the crista membranes and whether the level was similar to WT. I think this would be a very nice addition to the manuscript and make the story complex.

We thank the reviewer for raising this issue. However, the BN-PAGE analyses used here (Figure EV 1) do not allow us to draw conclusions on the sub-supercomplex organization. Based on our data, we feel that any statements on the different supercomplex forms could be questioned for its validity. Such types of analyses would require the use of low pore BN-PAGE analyses. However, this would be well beyond the scope and the aims of this study.

To address the point raised by the reviewer 1 and also to address reviewer 2 in an adequate and scientifically correct manner, we altered the text.

It now reads (page 7):

“Even in the absence of Mic60, virtually resulting in the absence of MICOS, the assembly of complexes I, II, and V was only somewhat impaired and the assembly of complexes III and IV was slightly decreased. This was also apparent for the corresponding supercomplexes (Fig EV1D).”

There were a few minor mistakes:

Results 4th line, should the authors also refer to fig 1C?

Page 9 first line. C is missing from Mic60

Page 11 last line, "they" should be clearly stated as to what the authors mean by they.

Page 16 Discussion, 5 lines from bottom of page. The sentence should read "allow us to draw conclusions"

Done. The textual changes were made accordingly.

Overall the manuscript is very nice and the work of exceptional high quality which should be published in EMBO.

Thank you.

Referee #2:

As previously noted, I am very excited by the quality and conclusions of this manuscript. I am happy for it accepted although I do request a slight change:

I am not keen on the revised sentence on page 7 :

"Even in the absence of Mic60, virtually resulting in the absence of MICOS, the assembly of complexes I, II, and V was nearly unaffected and the assembly of complexes III and IV was only slightly decreased (Fig EV1D)."

Rather than "nearly unaffected", "somewhat impaired" or similar would be more appropriate. In particular there does seem to be clear changes in the complex III-complex IV super complex profile in the Mic60 "knockout". This should not be dismissed in toto.

The textual changes were made accordingly.

It now reads (page 7):

"Even in the absence of Mic60, virtually resulting in the absence of MICOS, the assembly of complexes I, II, and V was only somewhat impaired and the assembly of complexes III and IV was slightly decreased. This was also apparent for the corresponding supercomplexes (Fig EV1D)."

Referee #3:

Authors performed an extensive revision of their paper and addressed experimentally and satisfactorily all the points raised.

Thank you.

The only missing experiment is the BN-PAGE for the MICOS holocomplex. From the co-IP it is impossible to conclude that the holocomplex is assembled (co-IPs are by definition performed on solubilized material), hence I suggest that before the paper is printed, this conclusion is modified.

Here, we disagree with the referee. BN-PAGE analyses represent a much harsher mode of analyses than immunisolations. They are carried out at exactly the same detergent concentrations as used for the immunoisolation. However, work by Schagger and von Jagow has shown that in the presence of Coomassie, non-ionic detergents display the properties of ionic detergents, thereby affecting complex integrity. Ample studies on mitochondrial protein complexes have shown that protein complexes are maintained by immunoisolation but destroyed by BN-PAGE. We would also claim that no study has demonstrated that any of the MICOS complexes resolved by BN-PAGE represents the MICOS holo complex, due to the arguments presented above.

Accepted

13th May 2020

I am pleased to inform you that your manuscript has been accepted for publication in The EMBO Journal.

Corresponding Author Name: Stefan Jakobs

Journal Submitted to: EMBO J

Manuscript Number: EMBOJ-2019-104105